# Influence of pump laser fluence on ultrafast myoglobin structural dynamics

Thomas R. M. Barends[1✉], Alexander Gorel[1,10], Swarnendu Bhattacharyya[2,10], Giorgio Schirò[3], Camila Bacellar[4], Claudio Cirelli[4], Jacques-Philippe Colletier[3], Lutz Foucar[1], Marie Luise Grünbein[1], Elisabeth Hartmann[1], Mario Hilpert[1], James M. Holton[5], Philip J. M. Johnson[4], Marco Kloos[6], Gregor Knopp[4], Bogdan Marekha[7], Karol Nass[4], Gabriela Nass Kovacs[1], Dmitry Ozerov[4], Miriam Stricker[8], Martin Weik[3], R. Bruce Doak[1], Robert L. Shoeman[1], Christopher J. Milne[4], Miquel Huix-Rotllant[2✉], Marco Cammarata[9] & Ilme Schlichting[1✉]

High-intensity femtosecond pulses from an X-ray free-electron laser enable pump–probe experiments for the investigation of electronic and nuclear changes during light-induced reactions. On timescales ranging from femtoseconds to milliseconds and for a variety of biological systems, time-resolved serial femtosecond crystallography (TR-SFX) has provided detailed structural data for light-induced isomerization, breakage or formation of chemical bonds and electron transfer[1,2]. However, all ultrafast TR-SFX studies to date have employed such high pump laser energies that nominally several photons were absorbed per chromophore[3–17]. As multiphoton absorption may force the protein response into non-physiological pathways, it is of great concern[18,19] whether this experimental approach[20] allows valid conclusions to be drawn vis-à-vis biologically relevant single-photon-induced reactions[18,19]. Here we describe ultrafast pump–probe SFX experiments on the photodissociation of carboxymyoglobin, showing that different pump laser fluences yield markedly different results. In particular, the dynamics of structural changes and observed indicators of the mechanistically important coherent oscillations of the Fe–CO bond distance (predicted by recent quantum wavepacket dynamics[21]) are seen to depend strongly on pump laser energy, in line with quantum chemical analysis. Our results confirm both the feasibility and necessity of performing ultrafast TR-SFX pump–probe experiments in the linear photoexcitation regime. We consider this to be a starting point for reassessing both the design and the interpretation of ultrafast TR-SFX pump–probe experiments[20] such that mechanistically relevant insight emerges.

Light is an important environmental variable and organisms have evolved a variety of photosensory proteins to sense it, exploit it, avoid it and deal with its damaging effects on, for example, DNA. Critical steps upon photon absorption include the formation of a photoexcited chromophore, coupled electronically and vibrationally to the protein matrix, resulting in a highly specific dynamic response taking the protein through a series of intermediates. The elucidation of these dynamic events is not only of interest from a basic scientific point of view but is also of practical significance. Many photosensory proteins are either medically relevant (visual rhodopsins, melanopsins and cryptochromes), are useful tools for cell biology (imaging via fluorescent proteins, functional manipulations in optogenetics) or are important for agriculture (photosystems and phytochromes). It is thus of great interest to understand the relevant chemical mechanisms (including molecular determinants of quantum yields), the different photophysical and photochemical pathways and the origin of structural changes that accompany and effect biological function.

Until recently, experimental investigations of ultrafast events following photoexcitation were limited to various optical spectroscopies. Such studies provide deep insight into electronic and vibrational changes during the reaction but only restricted structural information, thereby limiting mechanistic insight. This shortcoming has been alleviated with the advent of X-ray free-electron lasers (XFELs), which provide highly intense short X-ray pulses that enable ultrafast time-resolved serial femtosecond crystallography (TR-SFX)[1,2]. Importantly, SFX allows the use of microcrystals. The high chromophore

[1]Max Planck Institute for Medical Research, Heidelberg, Germany. [2]Institut de Chimie Radicalaire, CNRS, Aix Marseille Univ, Marseille, France. [3]Institut de Biologie Structurale, Université Grenoble Alpes, CEA, CNRS, Grenoble, France. [4]Paul Scherrer Institute, Villigen, Switzerland. [5]Molecular Biophysics and Integrated Bioimaging Division, Lawrence Berkeley National Laboratory, Berkeley, CA, USA. [6]European XFEL GmbH, Schenefeld, Germany. [7]ENSL, CNRS, Laboratoire de Chimie UMR 5182, Lyon, France. [8]Department of Statistics, University of Oxford, Oxford, UK. [9]ESRF, Grenoble, France. [10]These authors contributed equally: Alexander Gorel, Swarnendu Bhattacharyya. ✉e-mail: Thomas.Barends@mpimf-heidelberg.mpg.de; miquel.huixrotllant@univ-amu.fr; ilme.schlichting@mpimf-heidelberg.mpg.de

concentration in crystals results in high optical densities, which can be countered experimentally only by reducing crystal size. This is obligatory for the efficient and well-defined initiation of photoexcitation reactions.

In time-resolved pump–probe SFX experiments, microcrystals are delivered into the XFEL beam using mostly liquid jets and diffraction data are collected at distinct time delays following a photo-exciting pump laser flash. Using femtosecond- to sub-picosecond-long laser flashes, this approach has been used to study isomerization reactions in photoactive yellow protein[4], fluorescent proteins[5,16,17], various rhodopsins[6,7,9,10,12,14] and phytochrome[8]; electron transfer reactions in a photosynthetic reaction centre[11] and photolyase[13,22,23]; photodecarboxylation[24] and photodissociation[3]. In all cases, a very high pump laser fluence was used to maximize the light-induced difference electron density signal[20]. As a result, when using the same cross sections for ground state and excited state absorption, significantly more than one photon is nominally absorbed by each chromophore. Such excitation conditions differ markedly from those used in spectroscopic investigations, which are performed in the linear photoexcitation regime, with generally much less than 0.5 photon per chromophore. Multiphoton artefacts are then avoided and only the biologically relevant single-photon reaction is probed. Consequently, there can be considerable doubt as to whether SFX and spectroscopic measurements actually probe the same reaction, thus questioning the mechanistic relevance of the SFX results[18,19,25]. Nevertheless, the SFX community has failed so far to reach consensus on appropriate photoexcitation conditions for time-resolved pump–probe experiments[20,26].

Photodissociation of carboxymyoglobin (MbCO) is a well-characterized model reaction that has implications in a wide range of fields, ranging from organometallic chemistry to protein dynamics. The reaction has been studied by numerous computational and experimental approaches including TR-SFX[3], with issues of high photoexcitation power density having been pointed out early on[27,28]. Here we examine the influence of the laser fluence on structural features of photoexcited MbCO derived from TR-SFX experiments. We show that the dynamics of structural changes differ and that indications for coherent oscillations of the Fe–CO bond distance predicted by recent quantum wavepacket dynamics[21] are absent when using high photoexcitation power. As inferred from quantum chemistry, these differences can be explained by the sequential absorption of two photons, resulting in a different photodissociation mechanism.

## Power titration, occupancies and refinement

Power titration is a useful tool for establishing the linear photoexcitation regime, that is, the regime in which the magnitude of the response signal—or, in case of crystallographic investigations, the occupancy of the light-induced state—increases linearly as a function of the incident laser energy density. Our first pump laser power titrations of the MbCO photodissociation reaction in solution employed optical spectroscopy. To this end, we determined the laser-on/laser-off difference absorption spectra (range of 550–770 nm) 10 ps after photoexcitation by a 532 nm laser pulse. We explored different energy densities and pulse durations, specifically, three pulse durations of 80 fs, 230 fs and 430 fs at energy densities ranging from approximately 1 to 90 mJ cm$^{-2}$ in the centre of the Gaussian beam. The results are shown in Extended Data Fig. 1. The photolysis yield shows a clear dependence on the energy and duration of the pump pulse, with longer pulses being more efficient (up to approximately 60% for the 430 fs pulse (Extended Data Fig. 1d)). At fluences above approximately 20 mJ cm$^{-2}$, the shape of the transient difference spectra deviates from that of the static deoxyMb-MbCO difference spectrum, with a peak growing at approximately 650 nm (Extended Data Fig. 1a–c). Although this peak complicates the estimation of the photolysis yield within the high-energy density regime, it is clear that the linear photoexcitation regime lies below 10 mJ cm$^{-2}$

(Extended Data Fig. 1d). This value might differ somewhat when photo-exciting a microcrystal suspension. Therefore, we used TR-SFX at a 10 ps time delay on Mb.CO microcrystals, yielding structures to 1.4 Å resolution (Methods and Supplementary Table 1), to explore the influence of laser power (laser fluence 6–101 mJ cm$^{-2}$, see Supplementary Table 2 for excitation parameters) on photolysis yield. Inspection of the $F_{obs}^{light} - F_{obs}^{dark}$ (these are the observed structure-factor amplitudes from light-exposed and dark crystals, respectively) difference electron density maps shows a clear change of the magnitude of the peaks associated with bound and photolysed CO, respectively, and the haem iron. At higher laser fluence, changes are also apparent in the protein and the porphyrin ring (Fig. 1a). Considering only the magnitude of the difference density as in previous TR-SFX studies[8,10,20,29,30], a laser fluence of 101 mJ cm$^{-2}$ appears preferable. However, a mere visual inspection of the difference density may be misleading[1] since two effects may contribute to the difference density: a rise in occupancy of the photo-dissociated state with increasing fluence as well as potential structural changes related to multiphoton absorption. In general, resolving this issue requires structural refinement, which necessitates a reliable estimate of the occupancy. Indeed, the occupancy is a crucial parameter as an underestimated occupancy of the photolysed state may result in exaggerated structural changes upon refinement. Conversely, an overestimated occupancy may result in underestimated structural changes. However, determining the occupancy proved much less straightforward than we had anticipated, as detailed in Supplementary Note 1. To benchmark different methods to (1) determine the occupancy of the photolysed state and to then (2) refine its structure, we used simulated diffraction data, calculated from varying mixtures of dark-state (MbCO) and photolysed (MbCO*) molecules. It is important to note that different methods to determine the photolysed occupancy yield different absolute values, but show the same trends for the dependence of the occupancies from fluence or sub-picosecond pump–probe time delays (Supplementary Note 1). As rationalized in Supplementary Note 1 we decided to determine the photolysed occupancy in a given dataset by using the heights of the peaks for the photolysed- and dark-state ligand obtained from mFo-DFc maps calculated from refined models without CO. We used multicopy refinement to obtain the structure of the photolysed state, as this method makes fewer assumptions than structure factor extrapolation, and performed better than refinement against extrapolated structure factors in simulations (Supplementary Note 1) for the system investigated here.

Using these approaches for structural analysis of the power titration data reveals that, for example, the fraction of photolysed CO (denoted CO* henceforth) does not increase linearly at higher fluence, but instead levels off at approximately 80% (Fig. 1b). The reason that the photolysis yield is limited to 80%, despite very high laser fluence, is that a fraction of the thin plate-shaped MbCO crystals has at least one dimension that exceeds the 1/e laser penetration depth (approximately 7 μm) (Extended Data Fig. 2, Supplementary Table 2, Supplementary Note 2 and Supplementary Figs. 12 and 13). Our previous investigation, using very high fluence excitation and smaller crystals, showed 100% photolysis[3]. Importantly, both observations demonstrate nonlinear effects.

In the single-photon excitation regime, increasing laser fluence raises the occupancy of the light-induced state, but does not affect the amplitude or nature of the structural or electronic changes.

The nonlinear increase in CO* occupancy with laser fluence (Fig. 1b) is a clear indication for nonlinear effects induced by multiphoton excitation. Moreover, difference-distance matrix plots appear to show structural differences as a function of laser fluence (Fig. 1d and Extended Data Fig. 3a), as does the analysis of the displacements of Cα atoms from the porphyrin nitrogen atoms (Extended Data Fig. 3b). This confirms that the influence of multiphoton excitation on structural changes is not always immediately obvious in difference Fourier maps and may demand careful structural analysis.

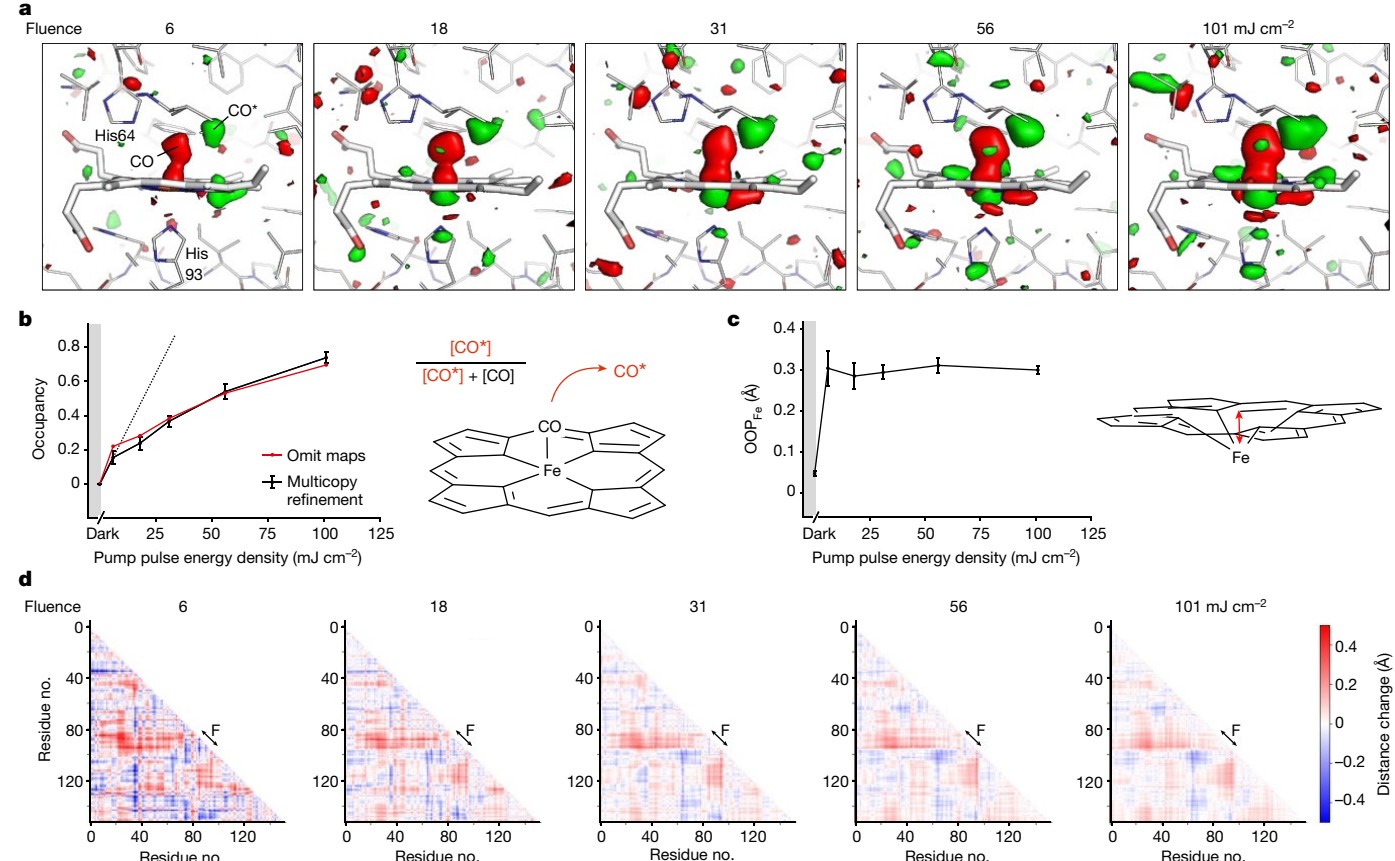

**Fig. 1 | Crystallographic power titration at 10 ps time delay.**
**a**, $F^{light}_{obs(\Delta t=10\ ps)} - F^{dark}_{obs}$ difference electron density maps (resolution 1.4 Å), contoured at +3.0 (green) and −3.0 (red) sigma, overlaid on the dark-state structure of myoglobin for the various pump energies. **b**, Apparent occupancy of the CO* state as a function of pump laser fluence (mJ cm$^{-2}$). The occupancy was determined in two ways: by dividing the CO* peak height by the sum of the CO* and dark-state CO peak heights in mFo-DFc omit maps (red line) and by refinement of mixtures of light- and dark-state models as described in the text (black line). **c**, Iron-out-of-plane distance as a function of pump energy (OOP$_{Fe}$). **d**, Cα–Cα-distance change matrices (Go-plots[45]) upon photoexcitation for the

various pump laser fluences (indicated above each plot). Red indicates an increase, blue a decrease in distance. The F-helix (indicated) containing the haem-coordinating His93 moves away from several other elements (B, C, D, E and G helices) and the E-helix moves towards the FG corner and the H helix. These distance changes between the F-helix and other structural elements are more pronounced at low fluence than at high fluence. Difference matrix plots between different pump laser fluences are shown in Extended Data Fig. 3. The error bars (**b,c**) are standard deviations determined by bootstrap resampling[46,47]. CO* denotes photodissociated CO.

## Structural changes at different fluences

We performed a multitime point pump–probe TR-SFX experiment on MbCO at SwissFEL to follow the CO photodissociation process at high temporal and spatial resolution to obtain insight into the reaction mechanism. To check whether the dynamics of the system are affected by the pump laser fluence, we used four different pump laser fluences (2.4, approx. 5, 23 and 101 mJ cm$^{-2}$, resulting in nominally approx. 0.3 to 12 absorbed photons/haem at the front of a crystal facing the pump laser beam, see Supplementary Table 2). These fluences are within, higher but still within, outside, and far outside the linear excitation regime, respectively (Fig. 1b). To increase the relative yield of photo-product at low laser fluence, we used smaller crystals for the 2.4 and approx. 5 mJ cm$^{-2}$ data series (Supplementary Table 2). The 2.4 mJ cm$^{-2}$ data show peaks at the expected positions for CO, CO* in the $F^{light}_{obs(\Delta t)} - F^{dark}_{obs}$ difference electron density maps (Extended Data Fig. 5). However, the signal-to-noise ratio of the data was too low to allow stable refinement (for discussion see Supplementary Note 1.3); these data will not be discussed further. The standard deviation of the time delays used in the SFX experiment is approximately 100 fs for the 5, 23 and 101 mJ cm$^{-2}$ data, taking into account timing jitter and the effects of data binning (Methods).

## Dynamics of MbCO photolysis reaction

The hallmarks of MbCO photolysis are the observation of an unbound CO accompanied by changes in the iron's spin states and position. Since CO photodissociates from the haem iron within 70 fs (ref. 31), and in line with our previous TR-SFX experiment[3] that showed full occupancy of CO* within the first time point, we did not anticipate any changes in CO* occupancy with time, given our time resolution. Unexpectedly, however, our electron density maps (including the initial $F^{light}_{obs(\Delta t)} - F^{dark}_{obs}$ difference maps, see Extended Data Fig. 6) show an apparent increase of the occupancy of CO* with time for the 5 and 23 mJ cm$^{-2}$ data and, to a lesser extent of the 101 mJ cm$^{-2}$ data (Fig. 2a). Since the data series were collected during two beam times using different batches of crystals, different dark-state data and different laser settings (Supplementary Table 2), it is very unlikely that this finding is a product of experimental errors. Of note, at 5 and 23 mJ cm$^{-2}$ the apparent increase of CO* occupancy occurs with time constants of approximately 350 and 450 fs, respectively, and are reminiscent of the damping constant of a coherent nuclear oscillation of CO* that was predicted by recent computational wavepacket analysis[21]. Since the time resolution of our experiment does not allow the predicted 1 Å amplitude, approximately 42 fs period oscillations to be resolved, they would manifest themselves

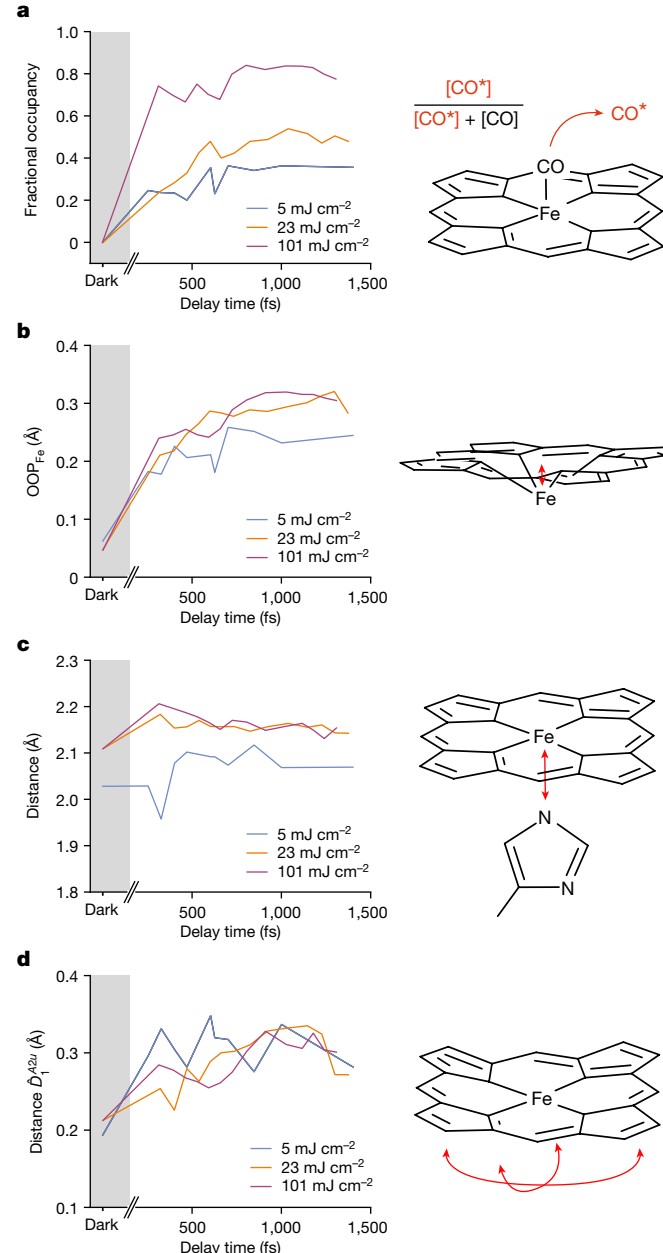

**Fig. 2 | Haem structural dynamics. a**, Apparent CO* occupancy (Fracc. occ.). Whereas at 5 and 23 mJ cm⁻² there is a smooth, slow increase, at 101 mJ cm⁻² there is a rapid initial rise, followed by an equally slow increase to the final amplitude. The 101 mJ cm⁻² curve can be understood as a superposition of contributions from the multiphoton-excited 'front end' of the crystals with the few-photon excited 'rear end' of the crystals (Supplementary Note 2 and Extended Data Fig. 2), resulting in almost instantaneous and apparently increasing occupancies of CO*, respectively. **b**, The iron-out-of-plane distance (OOP$_{FE}$) shows a larger amplitude with increasing fluence. **c**, The distance (Dist.) between haem iron and proximal His93 Nε2 atom, too shows differences between the fluences used, with the lowest fluence showing an oscillation and the highest fluence first going up and then settling at a lower amplitude. **d**, The haem doming ($\hat{D}_1^{A2u}$) also varies with the fluence, with again the lowest fluence showing an oscillation and the higher fluences do not. Estimates for the oscillation periods are indicated by red dashed lines in Extended Data Fig. 7. That figure also shows coordinate uncertainties.

simply as disorder due to distribution of the electron density over a large volume, resulting in an apparently low occupancy. As the oscillation damps, the CO* position 'narrows' and its apparent occupancy converges to the value observed for the respective laser fluences at

approximately 10 ps (Fig. 1b). According to this interpretation, at short time delays the crystallographic occupancy, which we determined from density peaks for the CO molecule, does not reflect the true yield of the photolysis reaction. Rather, the plateau value of the apparent crystallographic occupancy reflects the real photolysis yield. Accordingly, the time delay structures were all refined using the plateau value of the apparent crystallographic occupancy for the respective fluences (Methods and Supplementary Table 1).

Importantly, the predicted CO* oscillation seems to be suppressed in the high photoexcitation regime; our previous high fluence study showed maximal CO* signal within the first time delay[3]. Similarly, at 101 mJ cm⁻²—and in contrast to 5 and 23 mJ cm⁻² data—we observe an initial rise to about 85% of the final value within the first time delay of our experiment, then the final approximate 15% of the amplitude is reached with a speed comparable to what is observed at 5 and 23 mJ cm⁻² (Extended Data Fig. 2b). The time constants found for the 5 and 23 mJ cm⁻² cases, respectively, are not only similar to the decay of the computationally predicted oscillation, but are also comparable to a fast phase observed spectroscopically in single-photon excited hemoproteins[31,32] that is related to intersystem crossing.

We investigated the molecular basis for this experimental observation by quantum chemical analysis. As described previously[21], single-photon absorption by MbCO results in wavepacket transfer from the ground state to the singlet Q state of porphyrin, followed by transfer to the singlet metal-to-ligand charge-transfer (MLCT) band. The wavepacket undergoes large-amplitude coherent oscillations in the Fe−CO coordinate on the singlet MLCT band. Importantly, strong Jahn-Teller distortions in the excited state afford an efficient energy transfer from the porphyrin plane ($x, y$ polarization) to the Fe−CO axis ($z$ polarization), activating dissociative stretching vibrations and thus CO dissociation[21]. To assess the quality of the quantum chemistry results, we computed the iron-out-of-(haem)-plane (FeOOP) distance using molecular dynamics simulations in which a sudden dissociation of CO was imposed (Supplementary Note 3). While the simulations predict a larger final FeOOP distance than our experimental observations, the time constant with which the out-of-plane distance increases agrees well with our SFX observations (Extended Data Fig. 4 and Supplementary Note 3), and accordingly, we have high confidence in the accuracy of the computational approaches.

Our calculations suggest (Supplementary Note 3) that in the high excitation regime, CO dissociation occurs via a high-energy singlet state accessed by a sequential absorption of two photons. The first photon leads to the usual excited singlet Q state from which a second photon can be absorbed, as indicated by the absorption spectrum of the Q-excited haem-CO system (Extended Data Fig. 6). Analysis of the excitation character of this higher energy singlet state shows a mixed $\pi \to \pi^*$ character of the haem and $d_{xy} \to d_{z^2}/d_{yz} \to d_{z^2}$ character with respect to the ground state, and therefore is dissociative for the Fe−CO bond (Extended Data Fig. 6 and Supplementary Note 3). The potential energy surface of the singlet manifold along a relaxed scan coordinate at different fixed Fe−CO distances (Methods, 'Computational Details' section) clearly shows (Supplementary Fig. 11d) that, upon excitation to the dissociative singlet, after a second absorption from the Q state, the excited wavepacket experiences a rapid decay towards Fe−CO dissociation. This dissociation is thus driven by the sudden change in electronic structure induced by photon absorption. Due to the (barrierless) repulsive nature of the potential, no coherent oscillations of the wavepacket are expected to be observed, in contrast to the single-photon regime, in which nuclear motions drive the electronic structural changes that lead to dissociation. This explains the quasi-instantaneous initial increase in apparent occupancy of CO* in our high fluence TR-SFX data. In conclusion, the photophysical mechanism of CO dissociation differs for single and two-photon absorption, respectively, resulting in different structural dynamics of the dissociation process. This is in line with our experimental observations

obtained under the respective photoexcitation conditions which show large differences in temporal evolution of structural changes between the low- and high fluence regimes.

## Dynamics of haem and coordinating His93

Upon CO photodissociation, sequential changes of the Fe spin state occur, ultimately yielding the high spin state, and resulting in a movement of the iron-out-of-(haem)-plane (FeOOP) as well as motions of surrounding protein moieties. Here, too, our observations show marked differences between the single- and multiphoton excitation regimes. The plot of the temporal evolution of the FeOOP distance shows a strong increase within the first pump–probe time point, resulting in about 50% of the displacement, followed by a slower phase (time constant $\tau \approx 400$ fs) as reported previously[3,21,33] (Fig. 2b). Upon Fe movement, the Fe distances to the nitrogen atoms of the pyrrole ring (Np) and of the proximal histidine (His93), respectively, increase (Fig. 2c and Extended Data Fig. 7). In the 23 and 101 mJ cm$^{-2}$ data, the initially increasing Fe–His93 distance decreases again (Fig. 2c), possibly due to increased vibrational energy redistribution[34].

The haem dynamics upon CO photolysis have been studied by various spectroscopic methods, yielding time constants of processes and proposals for the structural basis of the underlying molecular changes. Our structural data are in line with the interpretation of X-ray absorption spectroscopy data by ref. 33 proposing changes of the FeOOP distance, the Fe–Np and Fe–His bonds with a time constant of 70 fs, followed by a smaller change of the FeOOP distance with a time constant of 400 fs. The latter was suggested to be linked to a movement of the F-helix, which we, however, observe on a 200–300 fs timescale depending on laser energy (see below). Our data do not agree with the structural interpretation in ref. 35, assigning a small FeOOP displacement to an 80 fs phase, followed by further FeOOP movement.

## Correlated protein structural changes

Oscillations of structural features (torsion angles, distances) of a light-sensitive cofactor and of nearby residues have been reported previously by TR-SFX[3,7]. These rapidly damped but coherent oscillations are a direct manifestation of the strong coupling of the chromophore and its environment. As in our previous study[3], we observe oscillatory dynamics in the haem environment, reflecting coherent motions excited by photodissociation in the haem (Extended Data Fig. 8). Importantly, the behaviour appears different for the various fluences[3].

Apart from these local dynamics, sequence displacement graphs[3,36]—which illustrate the change in distance of the protein main-chain atoms to the centre of the four porphyrin N atoms as a function of the time delay between the pump and probe pulses—show substantial main-chain changes within 1 ps throughout the whole protein for all pump laser fluences, but again the dynamics differ dramatically (Fig. 3a,b) between the fluence regimes. For many structural elements, the 5 and 23 mJ cm$^{-2}$ data display a temporal evolution over the entire ultrafast time series, whereas the 101 mJ cm$^{-2}$ data show essentially the entire displacement within the first time point, similar to our previous observation (Supplementary Fig. 5a in ref. 3). This is particularly noticeable for the displacement of the proximal His93 from the haem and the coupled motion of adjacent residues (Fig. 3). Moreover, a strong oscillatory modulation with a frequency of about 300 fs of the His93 displacement and to a lesser extent of the neighbouring residues (Fig. 3c) is clearly visible for the 5 mJ cm$^{-2}$ data only. Thus, the multiphoton effects are not limited to the small-scale motions of a few atoms but also affect larger-scale correlated protein motions in the entire protein (Extended Data Fig. 10), including the radius of gyration $R_g$ (Extended Data Fig. 10).

The striking change in dynamics of correlated motions (Fig. 3) with laser fluence is likely due to the excess energy deposited in the haem and Raman-active modes via multiphoton absorption, ultimately resulting

in heating[28]. At higher temperature, the displacement of the atoms from their equilibrium position increases so that modes sample more of the anharmonic part of the potential energy surface. As the rate of energy transfer between modes depends on the nonlinear coupling between them[37], they are then in effect more strongly coupled[28], resulting in faster structural changes. Moreover, our calculations suggest that sequential absorption of two photons of the same wavelength is possible. A purely repulsive higher excited state is reached with the second photon, leading to a ballistic Fe–CO dissociation, destroying the wavepacket coherent motion appearing in the single-photon process.

## Conclusions

The combination of spectroscopy, TR-SFX and quantum chemistry provides unprecedented insight into reaction mechanisms and protein dynamics, in particular when the initial ultrafast steps can be analysed as fully as only light-triggered reactions allow. An implicit assumption in such studies is that all three approaches study the same reaction, namely one triggered by the absorption of a single photon. Hence, photoexcitation conditions matter, in particular on the ultrafast timescale. For this reason, we repeated our previous[3] high fluence TR-SFX experiment on photodissociation of MbCO at lower fluence. Recent quantum dynamics computations have linked the microscopic origins of ligand photolysis and spin crossover reactions in photoexcited MbCO to nuclear vibrations and predicted coherent oscillations of the Fe–CO bond distance[21]. This prediction is consistent with our TR-SFX data showing an apparent increase of the CO* occupancy within 0.5 ps after low fluence photoexitation of MbCO, which mirrors the damping of the oscillation. In addition to providing experimental support of this computational prediction, our low fluence TR-SFX data also allow correlating of spectroscopically derived information[31–33,35] with structural data, including the coupling of modes[38]. Although our time resolution does not allow observation of the predicted coupling of the haem doming mode and the 220 cm$^{-1}$ (150 fs period) Fe-His mode[38], we observe the coupling of the FeOOP mode and the in-plane haem breathing mode.

High fluence excitation results in multiphoton absorption in MbCO. Our computations show that sequential two-photon excitation changes the photophysical mechanism by directly populating a dissociative state, bypassing the wavepacket oscillations and thus explain the distinct TR-SFX results under high fluence photoexcitation conditions. Moreover, apart from the difference in the resulting quantum state upon single- or multiphoton absorption, the latter also deposits more excess energy into the system, which opens further relaxation pathways because the thermal decay channel is strongly coupled to collective modes of protein[28,39]. It was shown previously[28] that under high excitation conditions MbCO displays power-dependent features with sub-picosecond components attributed to increased anharmonic coupling between the collective modes of the protein and the increased spatial dispersion of the larger amount of excess energy. Indeed, we observe faster and larger structural changes when using high fluence photoexcitation outside the linear regime (Fig. 3a,c). The changes are not purely isotropic but correlate with the energy flow; for example, the F-helix—which is directly linked to the haem via the proximal His93—is much more affected than the distal E-helix containing His64 (Fig. 3b). Moreover, the influence of the photoexcitation regime on oscillatory motions—which are much more pronounced in the low fluence data (5 mJ cm$^{-2}$, Fig. 3c)—complicates identification of coherent oscillations that are involved in mode coupling and ultimately result in the biologically relevant structural changes[40].

Time-resolved studies investigating reacting molecules typically aim to deduce their reaction mechanism by following the dynamics as the system evolves along a reaction coordinate in time. The different spectroscopic or structural probes trace electronic[41] and nuclear[40] changes and couplings[42] on the (sub)picosecond timescale that ultimately afford

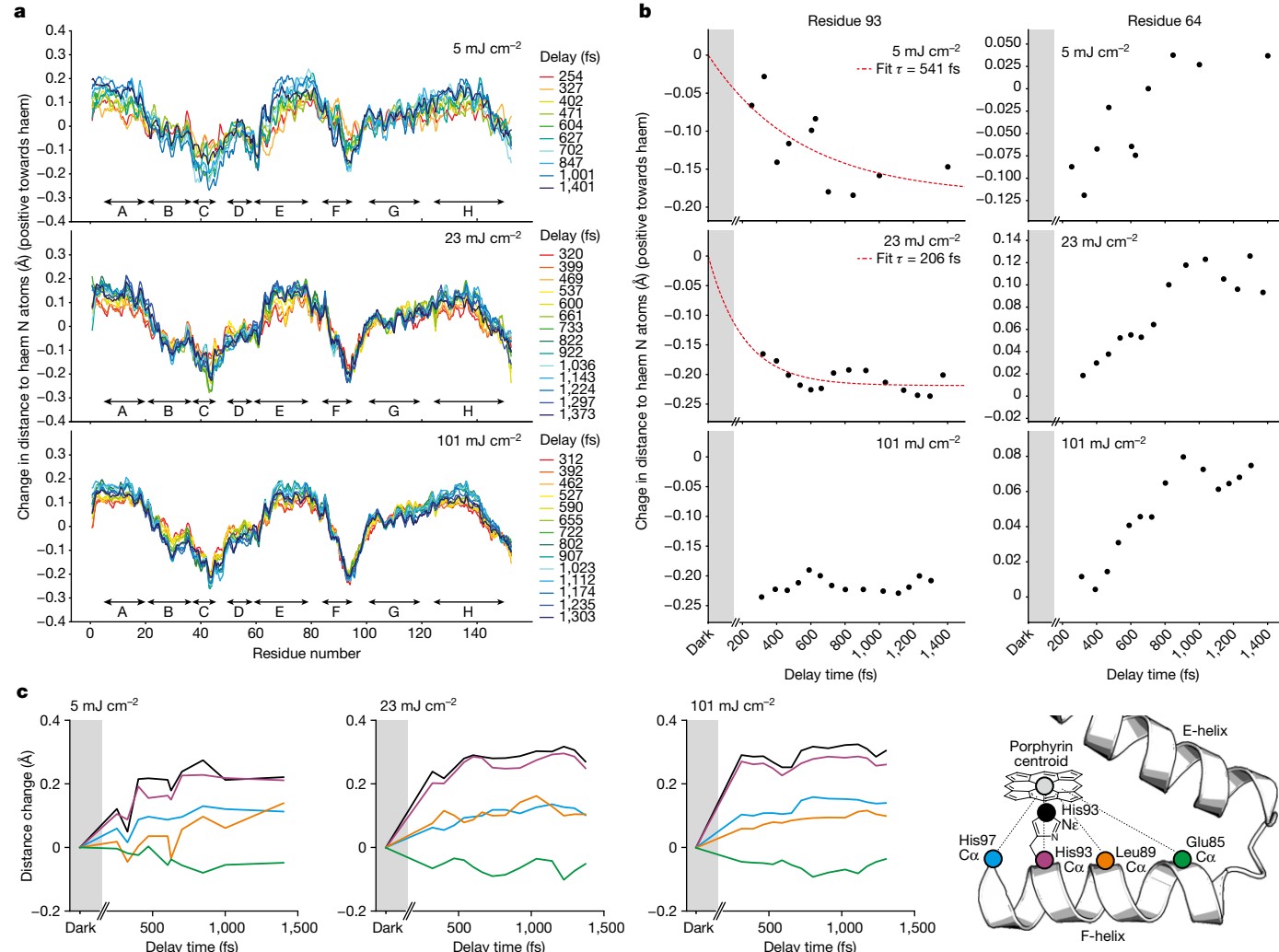

**Fig. 3 | Dynamics of correlated structural dynamics upon MbCO photolysis depends on laser fluence. a**, Guallar-type plots[36] showing the change in distance of backbone N, Cα and C atoms to the haem nitrogens for each time delay, for 5, 23 and 101 mJ cm⁻² pump pulse energy. The speed of the changes is strongly fluence dependent. **b,c**, Correlated motions of helical elements show different temporal evolutions with time in particular for the 5 mJ cm⁻² data, but move generally very fast in the 101 mJ cm⁻² data, obscuring the sequence of events. For example, the displacement of the His93 main chain from the haem nitrogen atoms or haem centroid has a time constant of $\tau \approx 540$ fs and $\tau \approx 210$ fs for the 5 and 23 mJ cm⁻² data, respectively, but reaches its final value within the first time delay for the 101 mJ cm⁻² data (**b**). In contrast, the movement of the distal His64 is hardly affected on the ultrafast timescale. **c**, The length of the correlated motion along the F-helix is clearly visible. Shown are displacements from the haem centroid of the His93 nitrogen (black) and Cα (purple), the Cα atoms of His97 (blue) and Leu89 (yellow) which are located at one helical turn upstream and downstream, respectively. Another turn further upstream (Glu85 (green)) the effect is strongly reduced. Importantly, a strong oscillatory modulation (period of about 300 fs) of elements of the F-helix (His97 Nε, black line, and His97 Cε, purple line) is only visible for the 5 mJ cm⁻² data.

formation of specific reaction intermediate(s). Thus, the characterization of the properties and structures of reaction intermediates is necessary but not sufficient for deducing the reaction mechanism. As a case in point, the crystal structure of the photolysed state of carboxymyoglobin has been known for a long time, but has provided little insight into the mechanism of photolysis and the structural dynamics of the system. This had been the realm of ultrafast spectroscopy until the advent of TR-SFX at XFELs. There is a great deal of spectroscopic evidence for many different systems showing that photoexcitation conditions matter, in particular when using femtosecond lasers, influencing yield (for example, via stimulated emission), magnitude and dynamics of changes as well as pathways taken (namely, mechanism), and species generated. Our current investigation demonstrates that this also applies to TR-SFX, which had been strongly debated[18–20]. Akin to the old adage that all roads lead to Rome, it seems quite possible that many systems reach long-lived intermediates irrespective of the reaction path taken, simply because the displaced modes are determined by the full molecular potential and are similar (but not identical) in terms of dynamical response, independent of whether activated by a single or by multiple photons[7,18].

Nevertheless, we show here that in investigating the road leading to Rome—that is, the ultrafast dynamics of structural changes, photophysical and photochemical mechanisms of the mechanistically relevant single-photon induced reaction—the situation differs: the temporal evolution of magnitudes of motions and spatial correlations vary depending on photolysis conditions used. Systems with excited state absorption in the spectral region of the pump pulse and excited state lifetimes of the pump pulse duration are particularly prone to multiphoton absorption through resonant processes[19]. This will be apparent from spectroscopic characterizations of the system investigated: such characterizations not only give the boundary conditions for the planned SFX experiment, but possibly also help identify pump wavelengths that yield excited states that are less prone to multiphoton absorption (shorter lifetime, shifted absorption), yet still show a

high quantum yield for the desired reaction and photoproduct[26,43]. Collecting more data instead of increasing the laser energy should be considered to avoid controversy[19] (see section 1.3 of Supplementary Note 1).

Given the widespread[19,20] use of overly high photoexcitation energies, it is likely that the ultrafast light-induced structural changes described for other systems that were presented and interpreted as mechanistically relevant for the single-photon reaction also involve multiphoton effects. Likely symptoms include large structural changes on the ultrafast timescale[8,11,14], including those referred to as protein quakes[6,44] and conformational transitions that are not in line with spectroscopic results[8,19,25]. Our results call into question recent statements promulgating the value of ultrafast TR-SFX pump–probe experiments performed above single-photon excitation thresholds[20].

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

## Methods

### Sample preparation

Horse heart myoglobin (hhMb) was purchased from Sigma Aldrich (M1882). After dissolving lyophilized hhMb powder (70 mg ml) in 0.1 M Tris HCl pH 8.0, the solution was degassed and then saturated with CO. Upon addition of sodium dithionite (12 mg ml$^{-1}$) while constantly bubbling with CO gas, the colour of solution turned to raspberry red. Dithionite was removed by desalting the protein solution via a PD10 column equilibrated with CO saturated 0.1 M Tris HCl pH 8.0. Subsequently, the MbCO solution was concentrated to approximately 6 mM using centrifugal filters before freezing in liquid nitrogen for storage.

hhMb crystals were grown in seeded batch by adding solid ammonium sulfate to a solution of 60 mg mM$^{-1}$ hhMB in 100 mM Tris HCl pH 8.0 until the protein started to precipitate (about 3.1 M NH$_3$SO$_4$). Seed stock solution was then added. Crystals appeared overnight and continued growing for about a week, yielding relatively large, often intergrown plate-shaped crystals[3]. Using a HPLC pump the crystalline slurry was fractured with tandem array stainless steel 1/4 inch diameter filters[48]. For beam time 1 (March experiment) the first tandem array contained 100 and 40 µm filters followed by a second tandem array of 40, 20, 10 and 10 µm filters. For beam time 2 (May experiment), the crystals were further fractured using a tandem array of 10, 5, 2 and 2 µm stainless steel 1/4 inch diameter filters. On average, the largest crystal dimensions of the crystallites were about 15 µm (Supplementary Fig. 12a) and about 9 µm (Supplementary Fig. 12b) for beam times 1 and 2, respectively.

### Laser power titration

Time-resolved spectroscopic data for estimating the extent of photolysis as a function of laser power density were obtained using a 6 mM hhMbCO solution. The sample was placed in a rectangular borosilicate glass tube sealed with wax to keep the solution CO saturated. The optical path length was 50 µm and the thickness of the glass tube was 1 mm. The optical density at the pump laser wavelength (532 nm) was about 0.5. An identical tube filled with the buffer solution (0.1 M Tris HCl pH 8.0) was used as a blank.

The fs laser pulses were generated by a Ti-sapphire amplifier (Legend, Coherent) seeded by a Mira fs oscillator. The laser output was divided into two branches: the vast majority was used as input of an optical parametric amplifier (Topas, LightConversion) to generate the pump pulses at 532 nm, while the remaining fraction was sent onto a sapphire crystal to generate short white-light pulses. Correction for white-light temporal chirp (of less than 2 ps over the probed window) was not needed at the time delay of interest. Mechanical choppers were used to lower the original 1 kHz repetition rate of both pump and probe pulses to 1 Hz and 500 Hz respectively. Pump and probe beams were spatially and temporally overlapped at the sample position and the relative time delay was set using a delay line. Pump pulses were focused to a full-width at half-maximum (FWHM) of about 0.1 mm, while the probing white-light FWHM beam size was about 0.02 mm diameter (FWHM). Each time-resolved spectrum was obtained by averaging 60 consecutive pump–probe events. A Berek compensator was used to change the pump light polarization from linear to circular. The 80 fs pump pulses were stretched to about 230 fs and 430 fs by inserting 10 and 20 cm water columns, respectively, along the pump laser path[49]. The difference spectra shown in Extended Data Fig. 1 were obtained using linearly polarized pump light; analogous results were found using circularly polarized light (data not shown).

### Data collection at SwissFEL

The TR-SFX experiment was performed in March (beam time 1)/May (beam time 2) 2019 using the Alvra Prime instrument at SwissFEL[50] (proposal no. 20181741). To follow the time-dependent light-induced dynamics, an optical pump, X-ray probe scheme was used. The repetition rate of the X-ray pulses was 50 Hz. Diffraction images were acquired at 50 Hz with a Jungfrau 16 M detector operating in 4 M mode. The outer panels were excluded to reduce the amount of data.

The X-ray pulses had a photon energy of 12 keV and a pulse energy of approximately 500 µJ. The X-ray spot size, focused by Kirkpatrick–Baez mirrors, was 4.9 × 6.4 µm$^2$ in March 2019 and 3.9 × 4.1 µm$^2$ in May 2019 (horizonal × vertical, FWHM). To reduce X-ray scattering, a beam stop was employed and the air in the sample chamber was pumped down to 100–200 mbar and substituted with helium. The protein crystals (10% (v/v) settled material, ref. 1) were introduced into the XFEL beam in a thin jet using a gas dynamic virtual nozzle (GDVN) injector[51]. The position of the sample jet was continuously adjusted to maximize the hit rate. In the interaction point, the XFEL beam intersected with a circularly polarized optical pump beam originating from an optical parametric amplifier producing laser pulses with 60 ± 5 fs duration (FWHM) and 530 ± 9 nm (FWHM) wavelength focal spots of 120 × 130 µm$^2$ and 150 × 120 µm$^2$ (horizontal × vertical, FWHM), in March and May, respectively. The laser energy was 0.5 and 1 µJ in May and 1–18 µJ in March 2019, corresponding to laser fluences of about 2.5 to 101 mJ cm$^{-2}$ and laser power densities of about 40 to 1,700 GW cm$^{-2}$ (Supplementary Table 2). Using an absorption coefficient of 11,600 M$^{-1}$ cm$^{-1}$ for horse heart carboxymyoglobin at 530 nm, this results in nominally approximately 0.3 to 12 absorbed photons/haem at the front of a crystal facing the pump laser beam. Time zero was determined in the pumped-down chamber at the same low-pressure helium atmosphere used for data collection. Information from a THz timing tool was used for determining the actual time delay. A power titration was performed at a 10 ps time delay (March 2019). Full time series were collected for pump laser fluences of 5 (May), 23 and 101 mJ cm$^{-2}$ (March). For the 5 mJ cm$^{-2}$ time series, the time delay could be set with sufficient reproducibility that each time point could be collected as a single dataset, with nominal time delays of $\Delta t$ = 150, 225, 300, 375, 450, 525, 600, 750, 900 and 1,300 fs. Using the timing tool available at the beam line, the actual time delays of these datasets could then be determined to be 254, 327, 402, 471, 627, 702, 847, 1,001 and 1,401 fs, with widths of approximately 85 fs. The number of indexed lattices in each dataset ranged from about 10,000 to greater than 30,000, and greater than 60,000 in the dark dataset.

At the time the 23 and 101 mJ cm$^{-2}$ time series were collected, the available timing reproducibility was less, and datasets were collected at a series of preset nominal time delays ranging from 150 to 1,300 fs that were then merged into large sets of about 150,000 indexed lattices for both fluences. These where then sorted according to the actual time delay of each image as determined by the timing tool of the beam line. Then, the data were split into smaller datasets by moving a window of 20,000 indexed lattices over the data for each fluence in steps of 10,000 indexed lattices. Thus, each of these datasets contain 20,000 indexed lattices, with an overlap of 10,000 indexed lattices between two consecutive time points. The timing distributions of these partial datasets have standard deviations of between 40 and 70 fs. In combination with the accuracy of the timing tool we estimate the true widths of these distributions to be approximately 100 fs. It should be noted that the overlap of the time delay distributions caused by this 'binning' of the 23 and 101 mJ cm$^{-2}$ data will result in a 'smearing out' of time-dependent effects.

The power titration data were collected during the same beam time as the 23 and 101 mJ cm$^{-2}$ data series, with the time delay set to nominally 10 ps. At this long time delay, the timing reproducibility of the beam line is of no concern and most heating effects have decayed. For the power titration, as many images were collected as was practical during the beam time, and the number of indexed lattices in each dataset varies.

Thus, while for the 23 and 101 mJ cm$^{-2}$ fluence time delay data, each dataset contains the same number of indexed lattices, whereas for the 5 mJ cm$^{-2}$ fluence time delay- and power titration data, there are different numbers of indexed lattices in each dataset. In serial crystallography, the precision of a dataset increases with the number of

indexed lattices. However, this should not affect the magnitude of structural changes beyond measurement error levels[52], and indeed, we observed no correlation of structural changes with the number of indexed lattices for the 5 cm² time delay data or the power titration data. Data statistics are given in Supplementary Table 1.

In each case, every 11th pulse of the pump laser was blocked, so that a series of ten light activated and one dark diffraction pattern were collected in sequence. High-quality dark datasets were generated by merging all laser-off patterns as well as separately collected, dedicated laser-off runs. The latter were also used to confirm that the interleaved dark data in the light runs were indeed dark and not illuminated accidentally.

## Diffraction data analysis

Diffraction data were processed using CrystFEL 0.8.0 (ref. 53); Bragg peaks were identified using the peakfinder8 algorithm and indexing was performed using XGANDALF[54], DIRAX[55], XDS[56] and MOSFLM[55,57]. Monte-Carlo integration[58,59] was used to obtain structure factor amplitudes. To calculate light-dark difference electron density maps, light data were scaled to the dark data using SCALEIT[60] from the CCP4 suite[61] using Wilson scaling. We investigated the use of different low- and high-resolution limits. Using a low-resolution limit of 30 Å worked for some datasets, but for others resulted in problems during light-dark scaling, likely due to differences in beam stop placement. However, we found that a low-resolution limit of 10.0 Å could be used for all datasets and this was therefore imposed for all calculations. Similarly, we found that a common high-resolution limit of 1.4 Å could be used for all photolysed structure determinations, which was implemented accordingly. The dark-state structures were refined against all available data.

Initially, occupancies of the photolysed state were determined by refining a model of the dark state without the CO ligand against the photolysed data and calculating mFo-DFc electron density maps using phases from a model. The heights of the peaks for the CO in the ground (dark) and photolysed CO* states were then used to calculate the occupancy $f$ using:

$$f = \frac{\rho_{CO*}}{\rho_{dark} + \rho_{CO*}}$$

where $\rho_{CO*}$ and $\rho_{dark}$ are the peak heights for the CO*- and dark-state CO peaks, respectively. These occupancies are shown as the red line in Fig. 1.

As is clear from the non-unity occupancies obtained, the structure factors originate from a mixture of the dark- and photolysed states. To obtain refined structures of the photolysed states, we considered refinement against extrapolated structure factors[62,63] as well as multicopy refinement. In this latter method, a mixture of the dark and photolysed states is refined against the original structure factor amplitudes. As multicopy refinement performed better than structure factor extrapolation in simulations (Supplementary Note 1), we continued with the multicopy refinement method. The occupancies were determined using a multicopy refinement-based approach that results in values that are very similar to the ones obtained using mFo-DFc omit maps (Supplementary Note 1).

For each photolysed structure, a starting structure for the light state was constructed from the appropriate dark-state structure, by moving the carbon monoxide molecule away from the haem and into the photolysed-state CO binding pocket. This photolysed-state starting structure was then combined with the appropriate dark-state structure to construct a range of dark/photolysed state 'mixture' pdb files with varying occupancies of the photolysed state (the occupancy of the dark state was set to 1-[photolysed state occupancy]). Each of these pdb files was then refined against the original photolysed data using phenix.refine build 1.19.2_4158 (ref. 64), allowing only the coordinates and B-factors of the photolysed state part of the mixture to vary. For all

refinements we used a haem geometry in which the planarity restraints were relaxed to allow the haem to respond to photolysis. The coordinates and B-factors of the dark state, as well as the occupancies of both states were kept at their starting values. After each refinement, the mFo-DFc difference electron density on the dark-state CO position was determined using phenix.map_value_at_point. At the correct occupancies of dark- and photolysed states, there should be no difference density at this position. The mFo-DFc densities at the dark-state CO position were then plotted against the occupancies of the respective mixtures. A line was fitted through these data points, and the occupancy at which this line crossed the $x$ axis (namely, where the mFo-DFc density at the dark-state CO position was zero) was taken as the correct photolysed-state occupancy for that particular dataset. These are the occupancies shown as the black line in Fig. 1 as well as in Fig. 2. A new mixture was then constructed with that occupancy for the photolysed state and refined in the same way (namely, while keeping the dark-state coordinates and B-factors as well as all occupancies at their starting values) to obtain the final, refined structure. As discussed in the main text, for the short time delay pump–probe data, the crystallographic occupancy, which we determined from density peaks for the CO molecule, does not reflect the true yield of the photolysis reaction. Rather, the plateau value of the apparent crystallographic occupancy is the real photolysis yield. Accordingly, the time delay structures were all refined using the plateau value of the apparent crystallographic occupancy for the respective fluences as the correct occupancy. Model statistics are given in Supplementary Table 1.

Structures were analysed using COOT[65,66], PYMOL[67] and custom-written python scripts using NumPy[68] and SciPy[69]. To obtain error estimates for structural parameters such as bond lengths and torsion angles, bootstrap resampling was performed as follows: of each dataset, about 100 resampled versions were created using a sample-and-replace algorithm. These were used to refine about 100 versions of each structure, which were used to determine standard deviations. The number of 100 resampled versions was chosen as this has been shown to result in sufficient sampling[46,47] while still being computationally tractable.

## Quantum chemistry

For the calculation of the absorption spectra and attachment–detachment density analysis, a reduced model in gas phase was constructed that includes the Fe-porphyrin along with CO on one side of the porphyrin plane and an imidazole (part of the proximal histidine) on the other side. The geometry was optimized at the DFT/B3LYP/LANL2DZ level. The absorption spectra were computed at the optimized singlet ground state geometry at XMS-CASPT2/CASSCF/ANO-RCC-VDZP level using OpenMolcas[70,71]. An active space of 10 electrons in 9 orbitals was used (5d orbitals of iron and 4π orbitals). The stick spectra were convoluted with Gaussians of 0.1 eV FWHM to obtain the spectral envelope.

For the relaxed scan along the Fe−C(O) dissociation coordinate, the geometries of the model system were optimized at fixed Fe−C(O) bond lengths on the lowest quintet ground state at the DFT level. XMS-CASPT2 calculations were performed at these geometries to obtain the PES cut, to extract 60 singlets included in the state-averaging to account for the dissociative state corresponding to the sequential two-photon absorption model.

The QM/MM model was constructed on the basis of the crystal structure of the horse heart myoglobin (PDB code 1DWR)[72]. The protein was solvated in a cubic box of 70.073 Å side length containing 11,684 water molecules. First, a minimization of the whole system was performed, followed by an NVT dynamics of 125 ps and a production run of 10 ns using Tinker v.8.2.1 (ref. 73). From the molecular dynamics (MD), we extracted several snapshots to perform quantum mechanics/molecular mechanics (QM/MM) MD, using a development version of GAMESS-US/Tinker[74]. The QM region includes the haem, CO and parts of the proximal- and distal histidines and was described at the

DFT level. The rest of the system is described at the MM level with the CHARMM36m (ref. 75) force field. A time step of 1 fs was used for the QM/MM molecular dynamics simulations.

## Reporting summary

Further information on research design is available in the Nature Portfolio Reporting Summary linked to this article.

## Data availability

Structures have been deposited with the PDB (accession codes 8BKH, 8BKN, 8R8F, 8R8G, 8R8H, 8R8I, 8R8J, 8R8W, 8W8X, 8R8Y, 8R8Z, 8R90, 8R91, 8R92, 8R93, 8R94, 8R95, 8R9C, 8R9D, 8R9E, 8R9F, 8R9G, 8R9H, 8R9I, 8R9J, 8R9K, 8R9L, 8R9M, 8R9N, 8R9P, 8R9Q, 8RA1, 8RA2, 8RA3, 8RA4, 8RA5, 8RA6, 8RA7, 8RA8, 8RA9, 8RAA, 8RAB, 8RAC, 8RAD, 8RAE); stream files, refinement and analysis scripts and relaxed haem geometry description with zenodo.org under https://doi.org/10.5281/zenodo.7341458. Diffraction images have been deposited in the CXIDB at https://doi.org/10.11577/2282689.

## Code availability

Analysis scripts are included in the zenodo archive under https://doi.org/10.5281/zenodo.7341458.

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

**Acknowledgements** We are grateful to M. Levantino for providing UV–Vis reference spectra of deoxy myoglobin and Mb.CO. S.B., M.H.-R and M.C. acknowledge the support by the French National Research Agency via the grant ANR-19-CE29-0018 (MULTICROSS). We thank J. Wray for critical reading of the text. We acknowledge support by the Max Planck Society.

**Author contributions** E.H., R.L.S. and I.S. prepared the samples. G.S. and M.C. performed the optical power titration. P.J.M.J., G.S. and M.C. carried out the laser work at SwissFEL. G.S., C.C., P.J.M.J., G.K., C.J.M. and M.C. operated the Alvra instrument at SwissFEL. T.R.M.B., A.G., G.S., C.C., J-P.C., L.F., M.L.G., M.H., P.J.M.J., M.K., G.K., K.N., G.N.K., D.O., M.S. M.W., R.B.D., R.L.S., C.J.M. and I.S. performed data collection at SwissFEL. M.K., M.L.G, M.S., G.N.K., R.L.S. and R.B.D. injected the crystals at SwissFEL. T.R.M.B., A.G., J.-P.C., L.F., M.H., K.N. and D.O. performed data analysis at SwissFEL. S.B. and M.H.-R. performed the quantum chemistry calculations. T.R.M.B., A.G. and M.H. performed off-line data analysis. Data were analysed by T.R.M.B. and I.S. J.M.H. contributed to crystallographic analysis; C.B. and B.M. provided spectroscopic input. T.R.M.B. and I.S. wrote the manuscript with input from all authors.

**Funding** Open access funding provided by Max Planck Society.

**Competing interests** The authors declare no competing interests.

**Additional information**
**Correspondence and requests for materials** should be addressed to Thomas R. M. Barends, Miquel Huix-Rotllant or Ilme Schlichting.

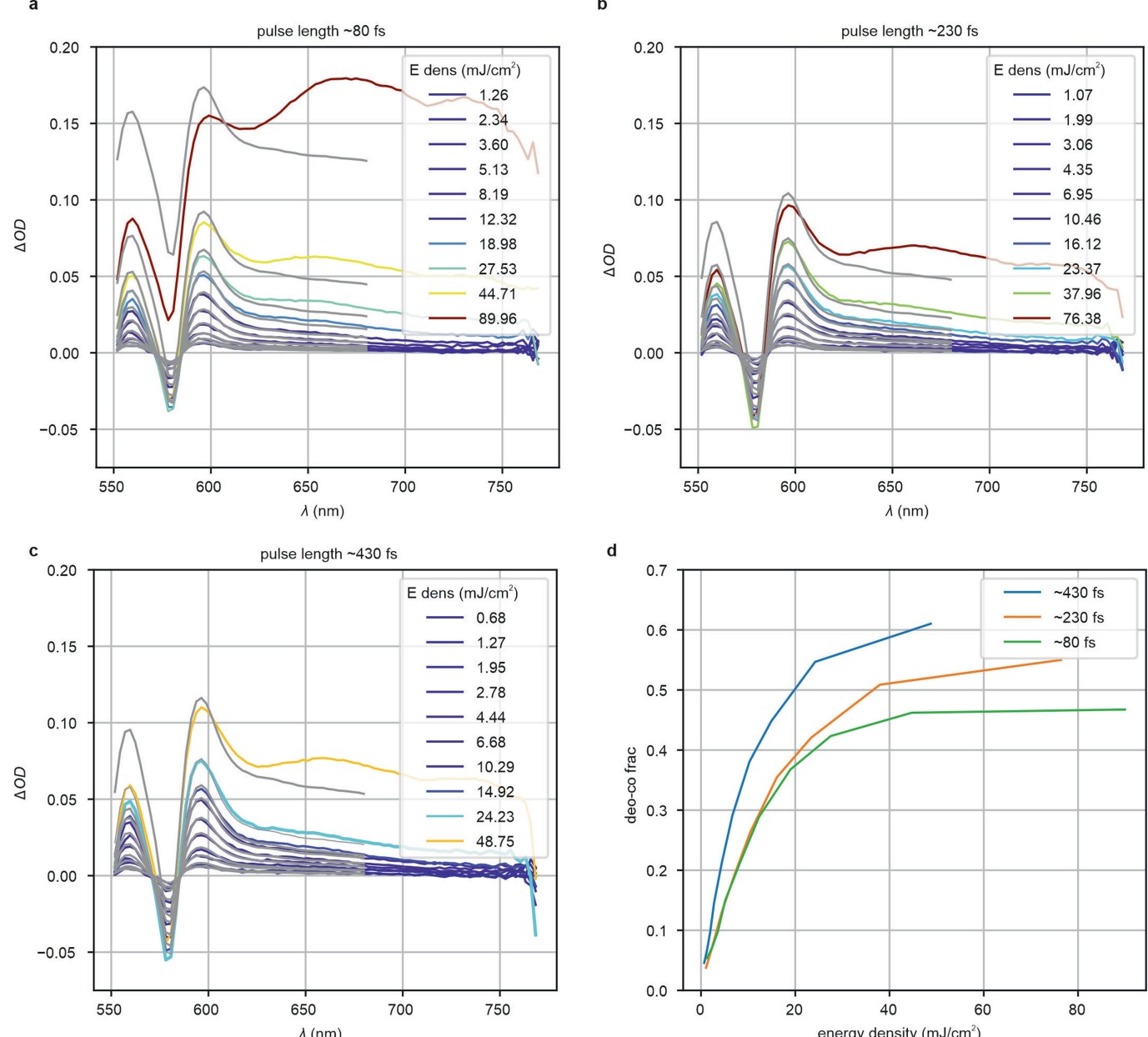

**Extended Data Fig. 1 | Optical power titration.** Carboxymyoglobin solution (6 mM, 0.5 OD at 532 nm) was photoexcited using three different pump laser durations (~80 fs (**a**), 230 fs (**b**), 430 fs, (**c**)) and different laser fluences, ranging from ~0.7 mJ/cm$^2$ to ~90 mJ/cm$^2$. Spectra were recorded after a 10 ps delay following a 532 nm laser pump pulse. The curves are colour-coded with respect to the energy density (fluence, same colour scale for **a**,**b**). The difference spectra, light-dark, were fit against difference spectra of deoxy myoglobin (the final state of the photodissociation reaction) and carboxymyoglobin (deoxyMb-MbCO + const. offset). The thin lines are the fits that were used to estimate the photolysis fraction shown in **d**. At high laser intensity the spectra change shape and an additional peak appears around 650 nm with a lifetime of a few ps (data not shown). The longer pulses seem to yield a higher fraction of photoproduct. **d**, Photolysed fraction calculated from the difference spectra. The traces are approximately linear up to 10 mJ/cm$^2$, allowing identification of the linear photoexcitation regime as ≤10 mJ/cm$^2$.

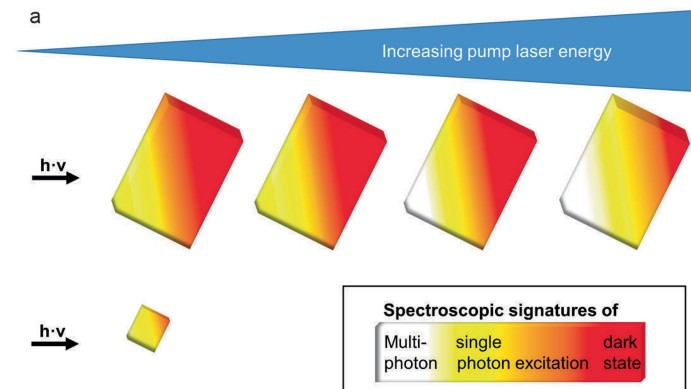

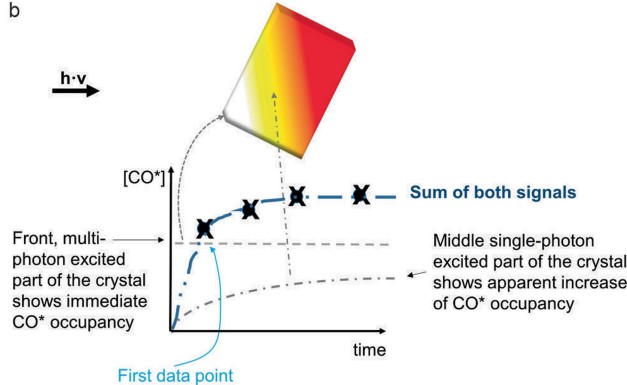

**Extended Data Fig. 2 | Crystal size, laser fluence and light-induced difference (light minus dark) signal are entangled quantities. a**, Low intensity laser light, such as required for excitation in the linear excitation regime, cannot traverse crystals that have a dimension that exceeds the 1/e laser penetration depth. When this dimension is parallel to the laser beam, a significant fraction of the crystal volume cannot be photoexcited and a large pedestal of dark molecules remains (red), resulting in small light-dark differences. To increase the signal, the laser fluence is increased, which however results in multiphoton absorption at the front of the crystal. The issue is much reduced for crystals that have thickness d < 1/e of the pump laser penetration depth[26]. **b**, The different photoexcitation conditions in relatively thick crystals at high laser fluence can reflect onto the signal. For example, in the 101 mJ/cm$^2$ fluence data the CO* signal increases strongly within the first time-delay, reflecting the very fast photodissociation upon multiphoton excitation at the "front" of the crystal. The increase of CO* with time is similar to the signal in the 5 mJ/cm$^2$ and 23 mJ/cm$^2$ data and originates from "deeper" regions in the crystal that were exposed to significantly lower fluence because of absorption by molecules in the "front".

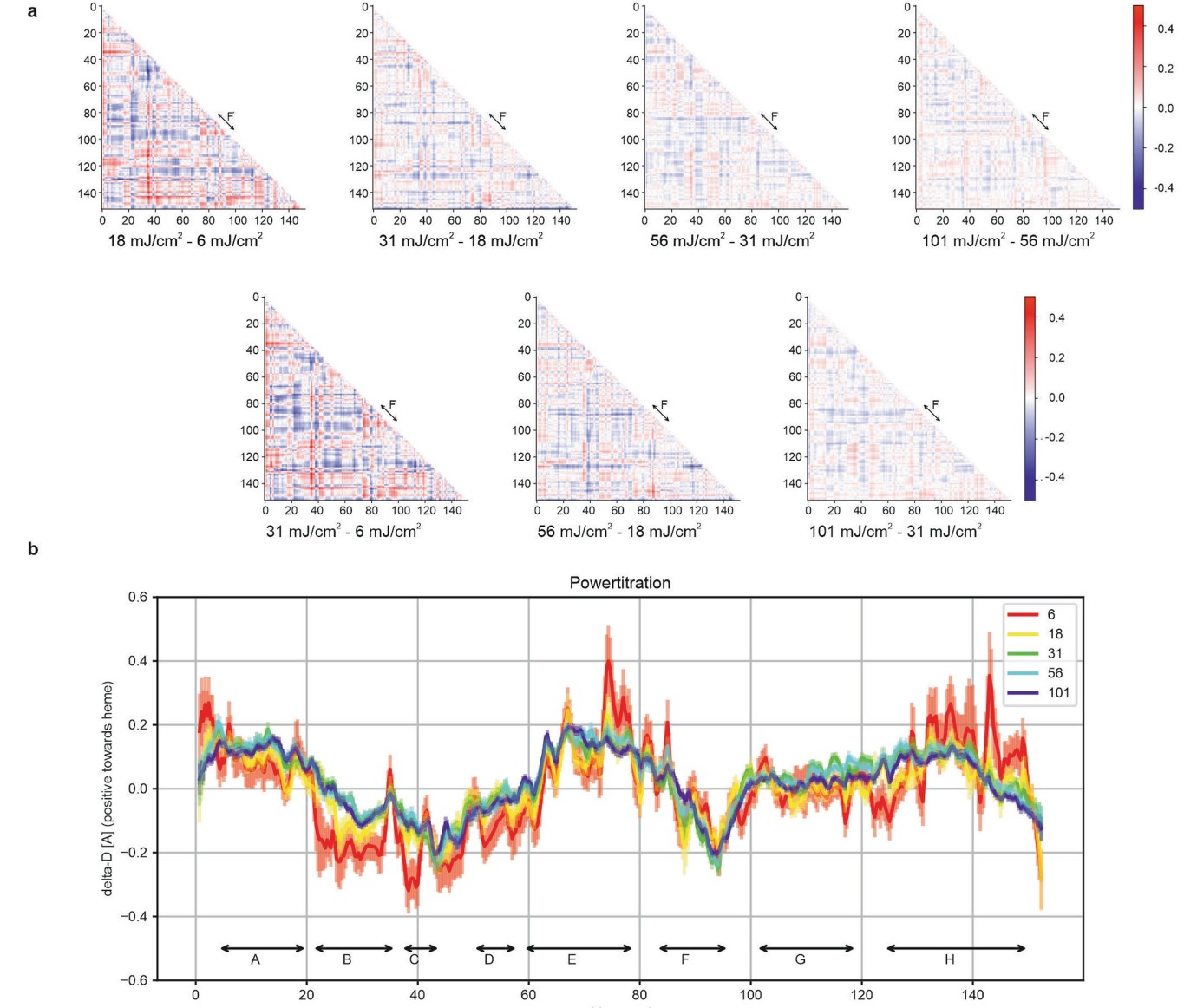

**Extended Data Fig. 3 | Structural changes as a function of pump laser fluence at a 10 ps time delay. a**, To facilitate the identification of systematic differences between the structural changes observed upon photoexcitation at different laser fluences (see Fig. 1d), we calculated difference difference plots. The red and blue colour-coding indicates that the atoms are further apart or closer together, respectively, than in the MbCO dark state structure. No systematic differences in correlated structural changes between pump laser energies are apparent. This differs from the findings described previously for a 3 ps time delay (see Fig. S13 in reference[3]) showing a more pronounced displacement for example of the F-helix at a pump laser energy of 20 μJ (-230 mJ/cm²; 1.5 TW/cm²) than at 6 μJ (-70 mJ/cm²; 450 GW/cm²). **b**, In contrast, Guallar-type plots[36], showing the change in distance of backbone N, Ca, and C atoms to the haem nitrogens for each time delay, show a clear difference for the different laser fluences. In particular, the changes are larger at lower fluence. Of note, the magnitude of the changes induced by 18 mJ/cm² photoexcitation is between those caused by lower and higher laser power density, respectively. Coordinate uncertainties (standard deviations derived from bootstrap resampling) are indicated (semi-transparent bars).

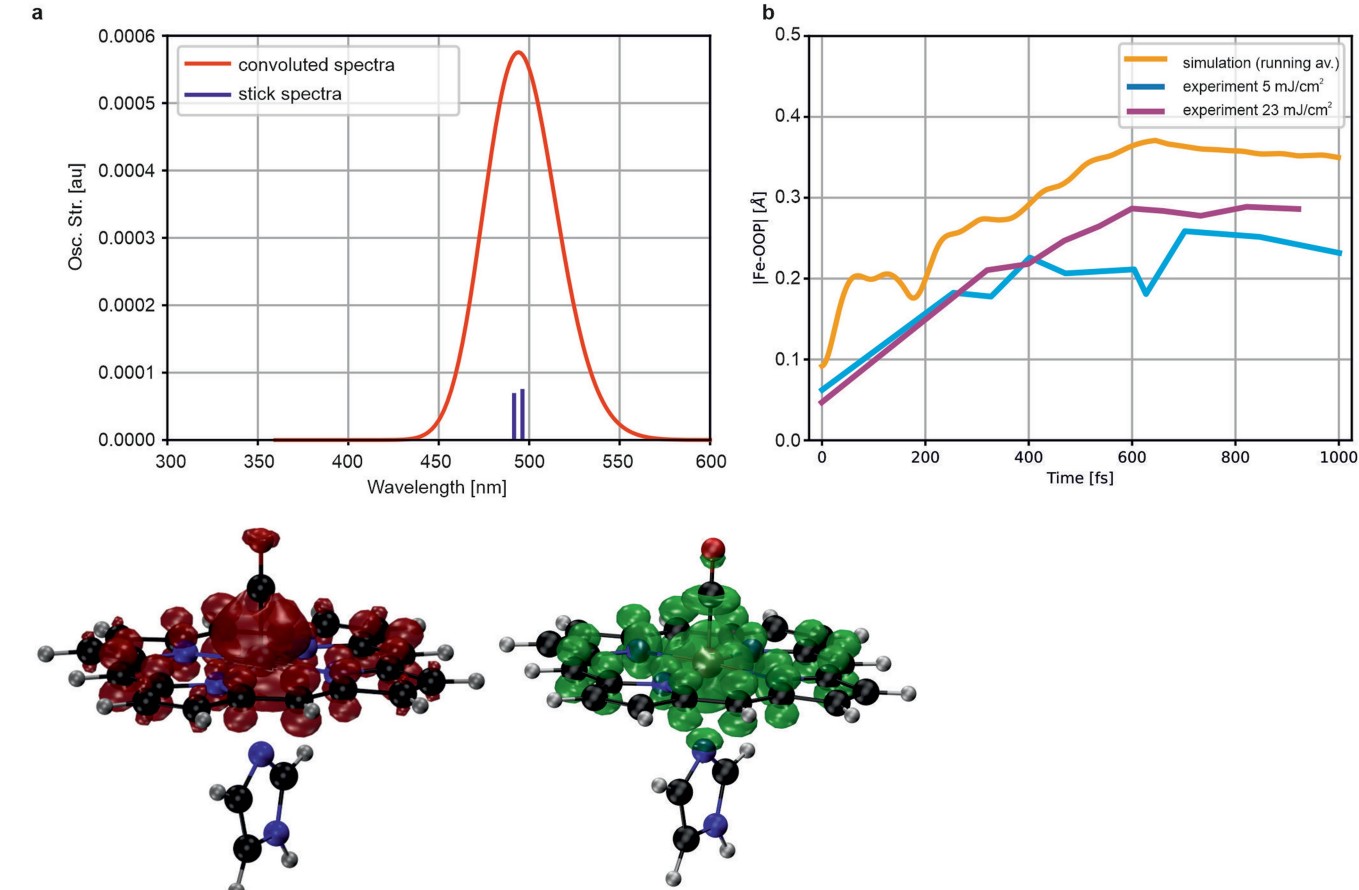

**Extended Data Fig. 4 | Quantum chemistry simulations. a**, The Q-band excited state absorption spectrum (top) can absorb to a high-energy singlet, in an excitation energy that is approximately two times that of the Q-band. This state corresponds to a mixed $\pi \to \pi^*$ character of the haem and $d_{xy} \to d_{z^2}$ / $d_{yz} \to d_{z^2}$ character with respect to the ground state, as analysed from an attachment (green)/detachment (red) density analysis, showing that it is dissociative with respect to the Fe-CO bond. **b**, Comparison of the FeOOP motion derived by QM/MM dynamics (see Supplementary Note 3 for details) and TR-SFX (single photon excitation, 5 mJ/cm² data, and 23 mJ/cm² data). The change in FeOOP distance over time predicted by the simulations shows a time constant that matches that observed experimentally at 5 and 23 mJ/cm². The reason for the difference in amplitude is not clear.

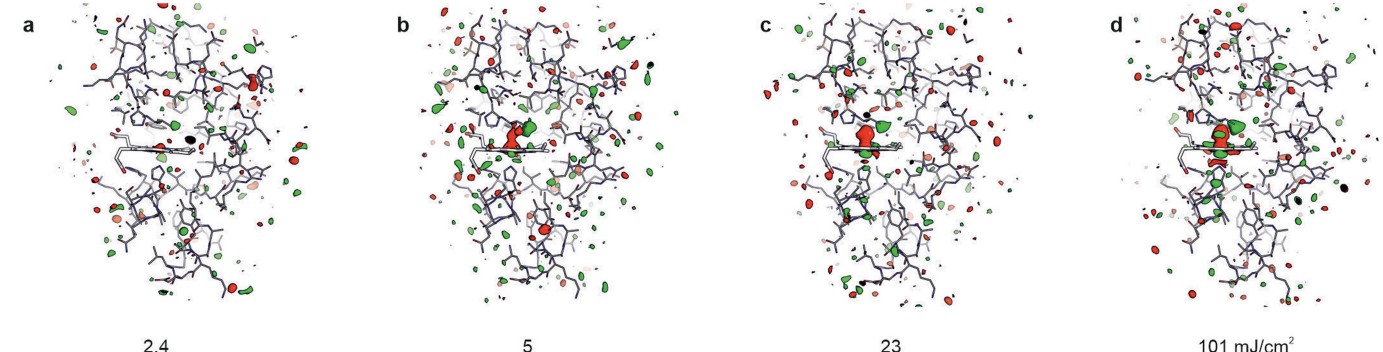

| a | b | c | d |
|---|---|---|---|
| 2.4 | 5 | 23 | 101 mJ/cm² |

**Extended Data Fig. 5 | Difference electron density maps, power titration.** "Long distance" view of $F^{light}_{obs(\Delta t \sim 1.3ps)} - F^{dark}_{obs}$ difference electron density maps (1.4 Å resolution) calculated from data collected from crystals photoexcited using laser fluences of 2.4 (**a**), ~5 (**b**), 23 (**c**) and 101 mJ/cm² (**d**). The maps are contoured at ±3 sigma, positive and negative peaks are shown in green and red, respectively.

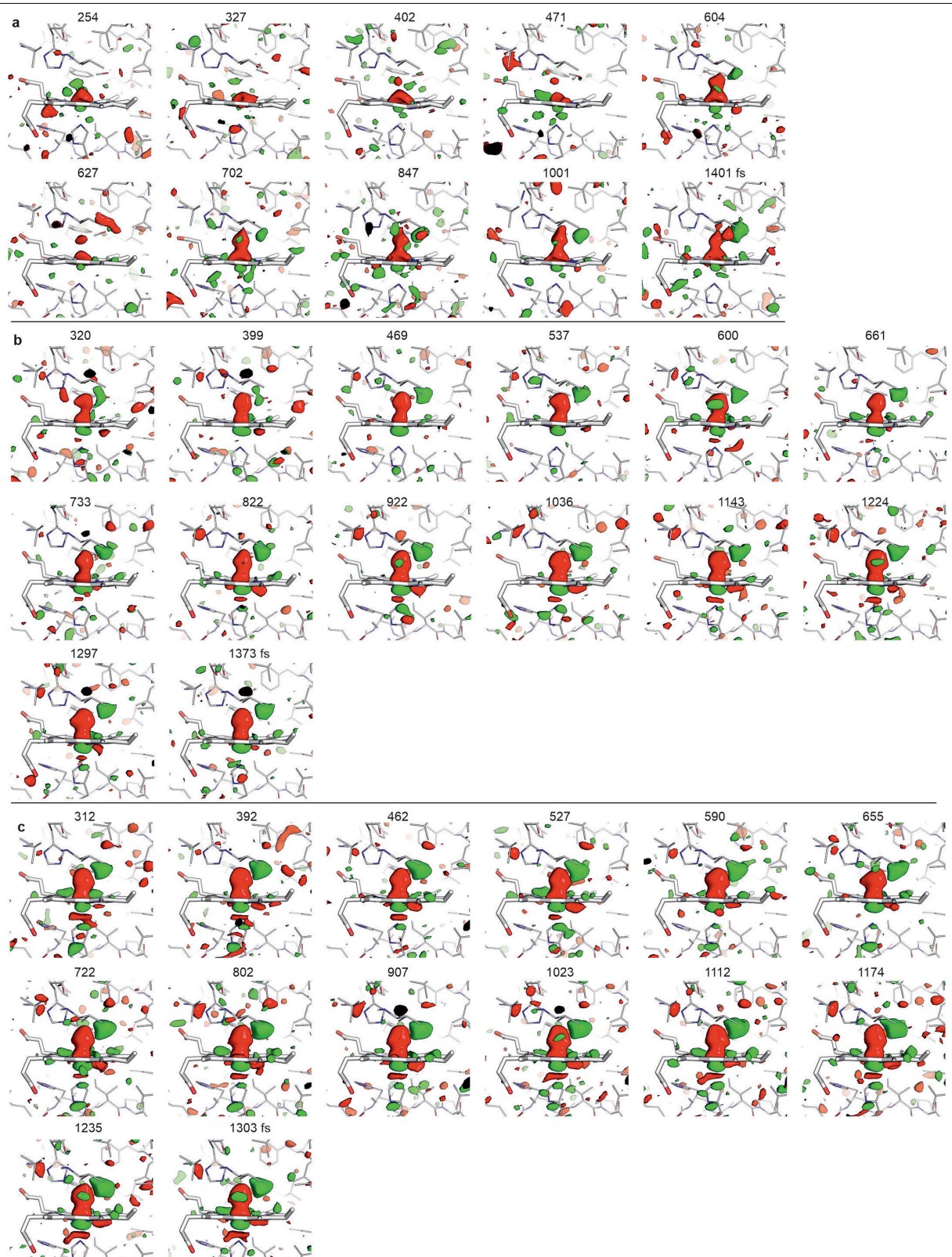

**Extended Data Fig. 6 | Difference electron density maps, time series.**
Temporal evolution of $F_{obs(\Delta t)}^{light} - F_{obs}^{dark}$ difference electron density peaks on the sub-picosecond time scale. As expected, the low fluence difference maps are much noisier than the high fluence difference maps. The intensity of the peaks corresponding to bound CO and photolysed CO*, respectively, hardly changes with pump probe delay time for the 101 mJ/cm² data (**c**). In contrast, the 5 and 23 mJ/cm² time series (**a**,**b**) show an increase of the magnitude of the CO* difference peak with delay time. Apart from the 5 mJ/cm² 627 fs time point data, which have very few indexed lattices, this increase does not correlate with the number of lattices included in each data set (indeed, the 23 mJ/cm² time points all have the same number of indexed lattices). The 1.4 Å resolution maps are contoured at ± 3 sigma; positive and negative peaks are shown in green and red, respectively. Delay times are indicated in fs.

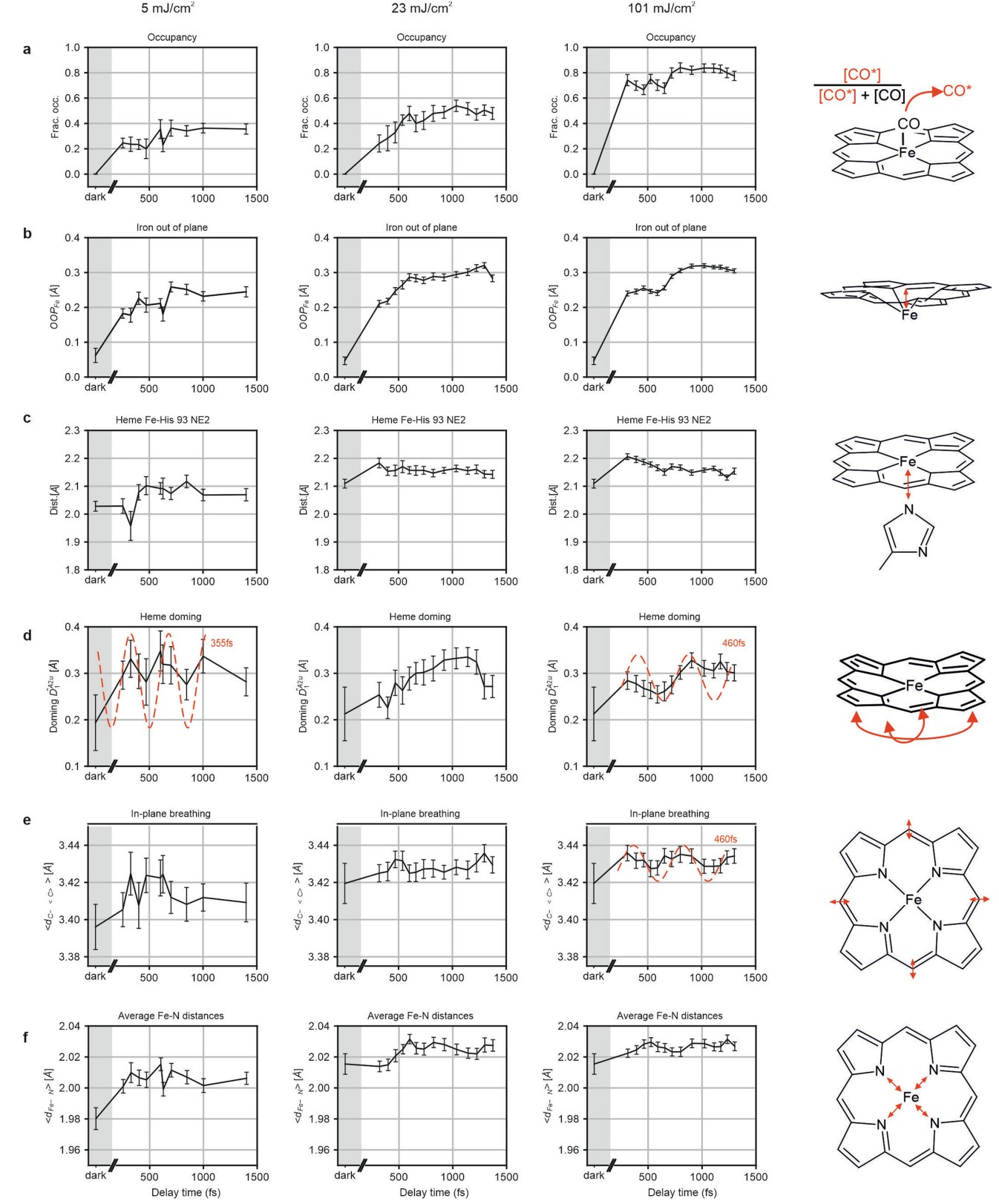

**Extended Data Fig. 7 | Haem structural dynamics.** The figure corresponds to Fig. 2 in the main text but shows more details. **a**, Apparent CO* occupancy; check the Fig. 2 legend for the temporal dependence of the 101 mJ/cm² data. **b**, Iron-out-of-plane distance; **c**, distance between haem iron and proximal His93 Nε2 atom; **d**, haem doming; **e**, haem in-plane breathing (v7 mode), determined as the average distance of the heme *meso* carbon atoms to the centre of the haem. **f**, Average distance between the iron atom and the porphyrin N atoms. The oscillation periods are indicated by red dashed lines. The coordinate uncertainties are indicated; they were determined using bootstrapping resampling as described previously[46,47]. Error bars correspond to ±1 sigma.

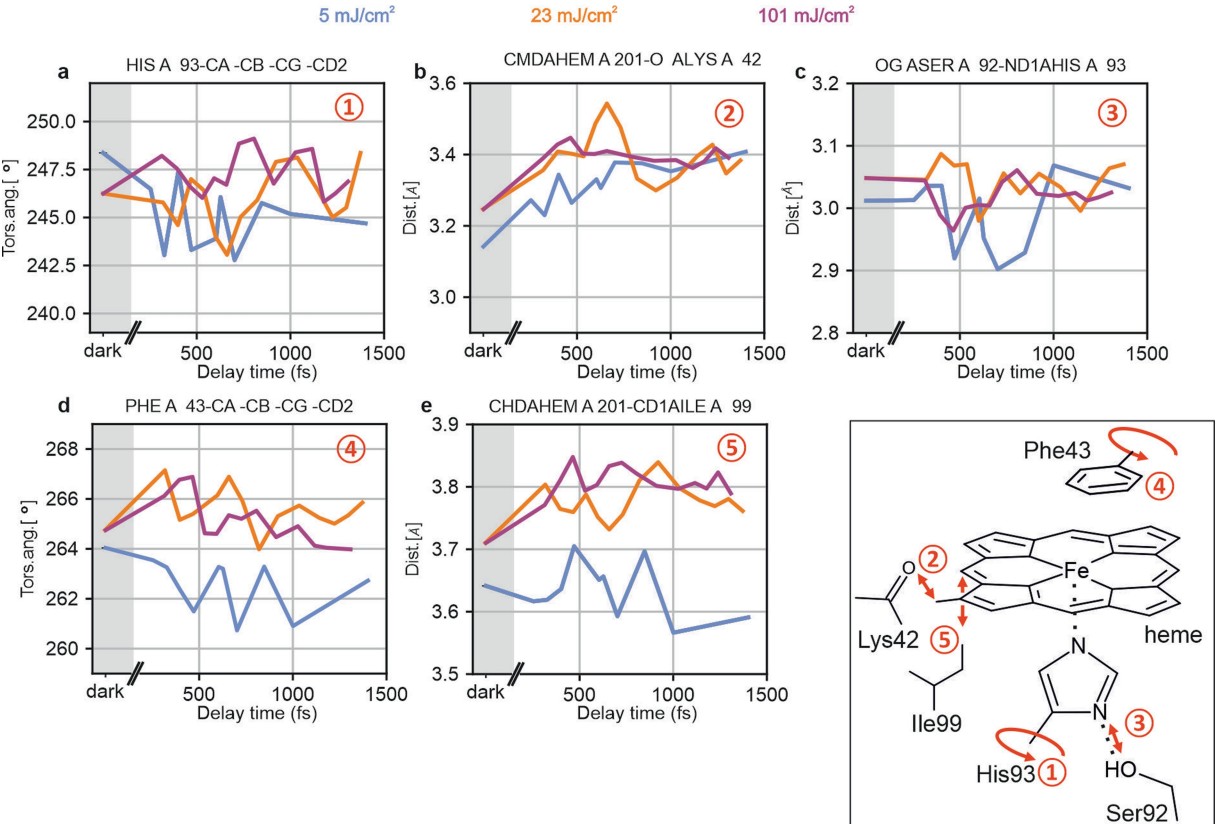

**Extended Data Fig. 8 | Dynamics of haem surroundings depend strongly on fluence. a**, χ2 torsion angle of the haem-coordinating His93. At higher fluences, much smaller movements are observed than at 5 mJ/cm². At 101 mJ/cm² an oscillation is observed that is not apparent at lower fluences. **b**, The distance between haem CMD atom and Lys42 backbone carbonyl O atom also shows different time evolutions with different fluences, with larger (and even oscillatory) motions at 5 mJ/cm² but smaller motions at higher fluences. Similar fluence-dependent effects are observed in **c**, the length of the His93 ND1…Ser92 OG hydrogen bond; **d**, the Phe43 χ2 torsion angle; and **e**, the haem CHD-Ile99 CD1 distance. Error bars corresponding to ±1 sigma as derived from bootstrap resampling and lines illustrating the oscillation periods are shown in Extended Data Fig. 9.

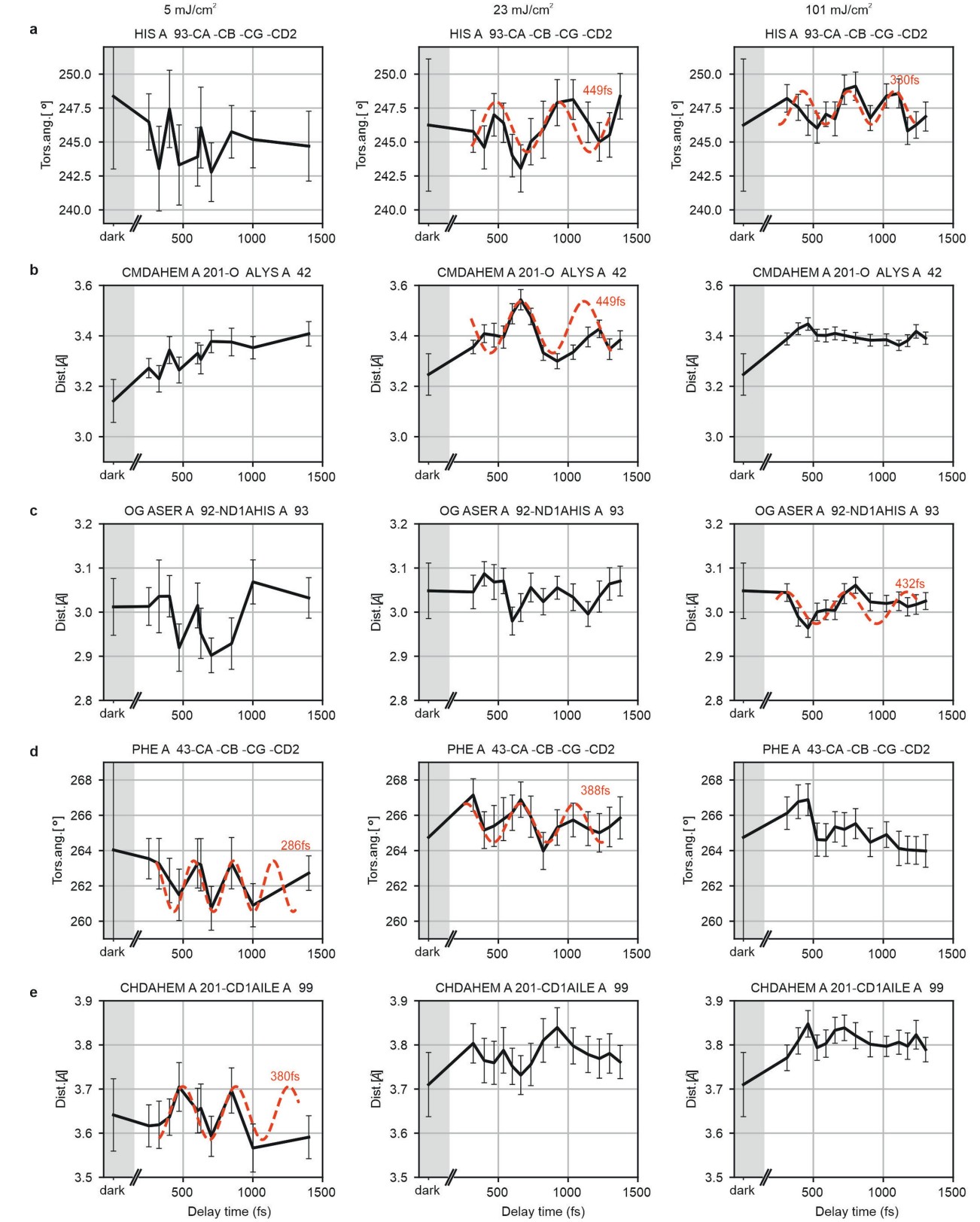

**Extended Data Fig. 9 | Dynamics of haem surroundings. a**, His93 χ2 torsion angle. **b**, Distance between haem CMD atom and Lys42 backbone carbonyl O atom. **c**, Length of the His93 ND1…Ser92 OG hydrogen bond. **d**, Phe43 χ2 torsion angle. **e**, Haem CHD-Ile99 CD1 distance. Red dashed lines illustrate oscillation periods. The coordinate uncertainties are indicated; they were determined using bootstrapping resampling as described previously[46,47]. Error bars correspond to ±1 sigma.

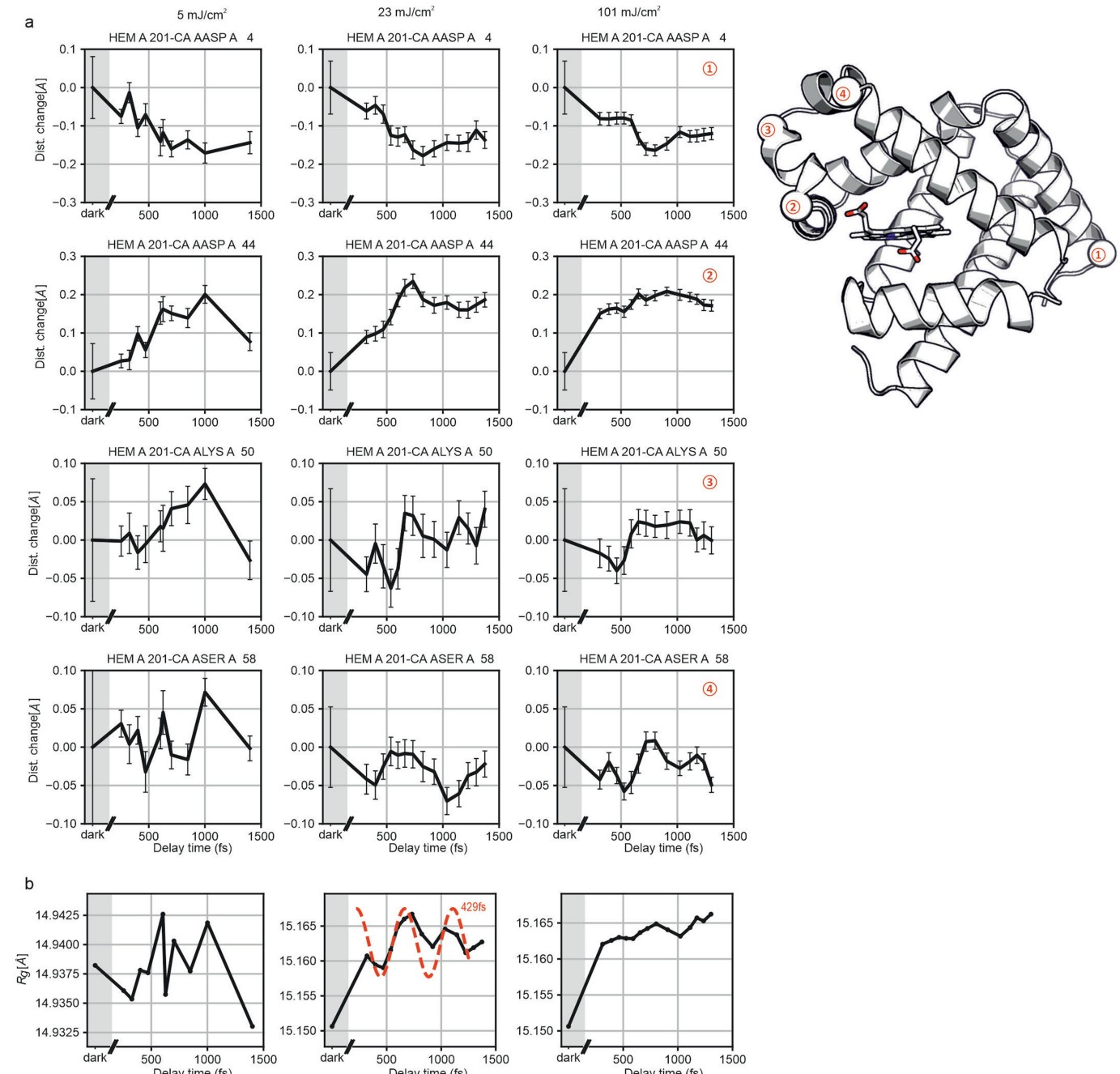

**Extended Data Fig. 10 | Cα atoms at the end of helices show an oscillatory modulation with time at low photoexcitation fluence.** Left panel 5 mJ/cm²; middle panel 23 mJ/cm²; right panel 101 mJ/cm². The location of the residues, chosen to be at the beginning or end of helices, is indicated. The F-helix (located below the haem, parallel to its plane) is shown in Fig. 3. The coordinate uncertainties are indicated; they were determined using bootstrapping resampling as described previously[46,47]. Error bars correspond to ±1 sigma. Differences in dynamics upon photolysis at different fluences are also apparent in the temporal evolution of the radius of gyration $R_g$ (**b**). Notably, the $R_g$ increases with delay time at fluences of 23 and 101 mJ/cm² but not in the 5 mJ/cm² single-photon regime.

# Reporting Summary

## Statistics

For all statistical analyses, confirm that the following items are present in the figure legend, table legend, main text, or Methods section.

| n/a | Confirmed | |
|---|---|---|
| ☐ | ☒ | The exact sample size (*n*) for each experimental group/condition, given as a discrete number and unit of measurement |
| ☒ | ☐ | A statement on whether measurements were taken from distinct samples or whether the same sample was measured repeatedly |
| ☒ | ☐ | The statistical test(s) used AND whether they are one- or two-sided *Only common tests should be described solely by name; describe more complex techniques in the Methods section.* |
| ☒ | ☐ | A description of all covariates tested |
| ☒ | ☐ | A description of any assumptions or corrections, such as tests of normality and adjustment for multiple comparisons |
| ☒ | ☐ | A full description of the statistical parameters including central tendency (e.g. means) or other basic estimates (e.g. regression coefficient) AND variation (e.g. standard deviation) or associated estimates of uncertainty (e.g. confidence intervals) |
| ☒ | ☐ | For null hypothesis testing, the test statistic (e.g. *F*, *t*, *r*) with confidence intervals, effect sizes, degrees of freedom and *P* value noted *Give P values as exact values whenever suitable.* |
| ☒ | ☐ | For Bayesian analysis, information on the choice of priors and Markov chain Monte Carlo settings |
| ☒ | ☐ | For hierarchical and complex designs, identification of the appropriate level for tests and full reporting of outcomes |
| ☒ | ☐ | Estimates of effect sizes (e.g. Cohen's *d*, Pearson's *r*), indicating how they were calculated |

*Our web collection on statistics for biologists contains articles on many of the points above.*

## Software and code

Policy information about availability of computer code

| Data collection | Data processing software is open source (CrystFEL) or freely available (CCP4 and PHENIX) |
|---|---|
| Data analysis | Analysis software is available from ZENODO |

For manuscripts utilizing custom algorithms or software that are central to the research but not yet described in published literature, software must be made available to editors and reviewers. We strongly encourage code deposition in a community repository (e.g. GitHub). See the Nature Portfolio guidelines for submitting code & software for further information.

## Data

Policy information about availability of data

All manuscripts must include a data availability statement. This statement should provide the following information, where applicable:
- Accession codes, unique identifiers, or web links for publicly available datasets
- A description of any restrictions on data availability
- For clinical datasets or third party data, please ensure that the statement adheres to our policy

| All data have been deposited with public databases. |
|---|

## Human research participants

Policy information about studies involving human research participants and Sex and Gender in Research.

| | |
|---|---|
| Reporting on sex and gender | n.a. |
| Population characteristics | n.a. |
| Recruitment | n.a. |
| Ethics oversight | n.a. |

Note that full information on the approval of the study protocol must also be provided in the manuscript.

# Field-specific reporting

Please select the one below that is the best fit for your research. If you are not sure, read the appropriate sections before making your selection.

☒ Life sciences ☐ Behavioural & social sciences ☐ Ecological, evolutionary & environmental sciences

For a reference copy of the document with all sections, see nature.com/documents/nr-reporting-summary-flat.pdf

# Life sciences study design

All studies must disclose on these points even when the disclosure is negative.

| | |
|---|---|
| Sample size | The number of images used for all datasets is indicated in the data statistics table, the rationale for bootstrap sample size choice is explained |
| Data exclusions | no data were excluded |
| Replication | Data were collected during two different beam times using two different batches of crystals. |
| Randomization | For cross-validation, reflections were selected randomly without possibility of influencing by the investigators |
| Blinding | Test set selection for cross validation was performed automatically, i.e. outside the control of the investigators |

# Reporting for specific materials, systems and methods

We require information from authors about some types of materials, experimental systems and methods used in many studies. Here, indicate whether each material, system or method listed is relevant to your study. If you are not sure if a list item applies to your research, read the appropriate section before selecting a response.

### Materials & experimental systems

| n/a | Involved in the study |
|---|---|
| ☒ ☐ | Antibodies |
| ☒ ☐ | Eukaryotic cell lines |
| ☒ ☐ | Palaeontology and archaeology |
| ☒ ☐ | Animals and other organisms |
| ☒ ☐ | Clinical data |
| ☒ ☐ | Dual use research of concern |

### Methods

| n/a | Involved in the study |
|---|---|
| ☒ ☐ | ChIP-seq |
| ☒ ☐ | Flow cytometry |
| ☒ ☐ | MRI-based neuroimaging |

