## [Peer Review File · Nature]

Manuscript Title: Influence of pump laser fluence on myoglobin ultrafast dynamics

Reviewer Comments & Author Rebuttals

Reviewer Reports on the Initial Version:

Referees' comments:

Referee #1 (Remarks to the Author):

The paper by Barends et al "Influence of pump laser fluence on ultrafast structural changes in myoglobin" is an unqualified breakthrough and should definitely be published in Nature. It is the first work to exploit the femtosecond time resolution possible with XFEL sources to directly observe the primary processes that couple chemical driving forces to biological functions – without multiphoton artifacts. The work was carefully conducted in the linear 1-photon excitation regime involving for the first time for TR-FSX studies of a well defined initial state for which we can directly connect to the biological process. In this case, the change in electron density associated with bond breaking at the iron site and the ensuing structural motions of the protein that are involved in directing ligation and ligand escape as part of ligand transport exhibited in heme proteins.

There may be some question about connecting a phototriggered bond breaking process to that of the thermally sampled reaction coordinate in terms of biological relevance, however, there are sufficient degrees of freedom involved that the coupled motions to the bond dissociation coordinate (thermal or phototriggered) should be within linear response and reflect the underlying mechanism involved in ligand dissociation and escape in understanding the overall process of ligand transport in biological systems. It is, however, essential, to get the initial state preparation correct to have any hope to correlate the observed structural dynamics to biologically relevant motions. In this respect, this work is a true breakthrough. The authors were able to improve the signal to noise for the desired 1-photon process by explicitly going to crystals with the same dimension as the 1-photon absorption depth. They also rigorously conducted essential optical control experiments to determine the peak power threshold for multiphoton artifacts. The data analysis has also been dramatically improved. The thinner crystals significantly reduce the x-ray diffraction efficiency or total x-ray counts, as the diffraction efficiency scales quadratically with thickness. The data collection protocols and analysis are really a tour de force in averaging diffraction patterns, proper binning, and careful power dependences to ensure the signal was collected under the biologically relevant 1-photon limit.

The authors were able to pull out a wealth of information on the protein response function that I can see will resolve a lot of open questions. To give some impression of the significance of this accomplishment, it is apparent from the review of R. Neutze in Science 2021 (see Table 1 in this review) that all previous experiments have been conducted at extremely high fluence and peak powers where multiphoton absorption processes dominate and lead to other reaction pathways than the biologically relevant 1-photon process. It was argued in this review that the high excitation conditions still access the biologically relevant information even if the motions at early time are exaggerated. The present submitted work clearly demonstrates that this assertion is not correct. Both the pathway, correlation lengths, and most important the amplitude of the motions, are significantly different in direct comparison of high excitation (multiphoton) to the low excitation (1-photon) biologically relevant excitation conditions.

This work finally allows us to gain insight into the correlation lengths and energy exchange among the different degrees of freedom involved in biological processes. This issue is one of the big mysteries in science in terms of how chemistry scales in complexity and is able to be connected to

biological macromolecules to execute functions - without getting trapped in local minima. At the heart of the issue is the surrounding protein matrix about an active site has an astronomical numbers of degenerate conformations and corresponding minima that otherwise be expected to block the process from occurring on the timescales needed for establishing connected biochemical pathways. This problem is similar to Levinthal's paradox in protein folding but in this case refers to fluctuations involved in the chemical active conformation subspace. An obvious solution is that not all the nuclear degrees of freedom are independent. There must be spatially extended correlations in nuclear motions that are coupled to the reaction coordinate to avoid this apparent conundrum. The collective mode coupling model was put forward to explain the basic physics involved but the open question was which particular modes and correlation lengths are involved. Put simply, every object has a k (length) and frequency (ω) dependence inherit to the collective forces involved. We don't know the length scale and time scales for the protein response function to the primary forces involved in chemical processes. The bond dissociation coordinate for the CO bond to myoglobin provides an excellent model system to probe this issue. The MbCO system is particularly important as it forms the basis for understanding the allosteric mechanism involved in molecular cooperativity where hemoglobin is the cornerstone. As apposed to other phototriggered biological processes, where the local changes to the active site control barrier heights and biological responses, the very function of heme proteins involves changes in the protein structure to communicate the state of ligation at adjacent hemes. The observed structural changes in this work are truly at the heart of the problem and constitute the first direct observation of functionally relevant protein motions.

I have carefully looked at all the data and the projection of the different nuclear spaces to observe the degree of correlations. I am extremely impressed by the degree of rigor in this work. In my opinion the most important figures are figure 3 showing the power dependence for the key F helix motion that has long been associated with coupling between protein subunits in directing allosteric regulation. The difference in specific motions and amplitude for the different powers definitely shows the importance of going to low enough excitation to be in the 1-photon limit to access these functionally relevant protein motions. Similarly, the extended figure 8 shows the dramatic difference in the helical motions surrounding the heme pocket. It is important to note that for specific details the difference in displacement is as much as a factor of 2 between data collected in the linear 1-photon regime and high power multiphoton regime. This difference represents a factor of 4 difference in reorganization energy (within linear response and quadratic scaling with displacement). This is not a minor difference, it would constitute a 2 orders of magnitude larger barrier to the back propagating ligation coordinate. These are not minor differences. Further, it is always possible to look for similarities and state the motions observed within the 1-photon limit and those driven by multiphoton processes are similar, as has been done for all other experiments published to date. Here the differences are both dramatic in terms of spatial correlations, locations of maximum strain (displacement) and critical magnitudes. It is the amplitude of the motions coupled to the interface that are crucial to understand in relation to the mechanisms of allosteric regulation. In this respect, the very last figure on the radius of gyration (extended figure 8b) makes a very clear statement. It is known from optical studies that the volume changes (related to the radius of gyration, R_g) in the first nanoseconds (Terazima et al, transient grating studies) are very small, as is observed for the excitation within the 1 photon limit. The multiphoton R_g is .2 Angstroms and clearly overdamped from excessive heating compared to the 1-photon excitation conditions where there is evidence for damped oscillations from the impulsive relaxation coupled to the bond breaking over the collective mode network of the protein. For excitation in the 1-photon limit, the net change is very small. This difference is huge and is critically related to the allosteric coordinate.

I recommend this paper for publication as is. The only minor point I can make is that extended figures 7 and 8 do not have the b panels labelled and the discussion is a bit confusing. Everything else is very clearly discussed and the level of detail impressive.

In summary, the paper constitutes a major advance to the field of time resolved x-ray diffraction.

It is the first ultrafast experiment to resolve the primary motions coupled to functionally relevant protein motions within the linear absorption, 1-photon, limit that is essential to provide a well-defined initial state to correlate the ensuing structural dynamics. Further, the observed structural dynamics map very well onto the known dynamics from all optical studies. The collective mode coupling mode for functionally relevant protein motions could be argued to have been demonstrated from all optical studies, however, the details in the helical motions and correlations lengths are now observable for the first time. This work also demonstrates by careful attention to creating samples with pathlengths within the $1/e$ absorption depth, it is possible to do this class of experiments. The authors have shown that not only is it possible – it is essential. This work now paves the way for revisiting the structural dynamics of other work and developing the protocols for this class of experiments to ensure the observed dynamics are biologically relevant. I strongly recommend publication in Nature. It will open the eyes of the entire community and put trust that this class of experiments can mine biologically relevant information.

This is beautiful work. The authors are to be congratulated. (It satisfies all the require criteria for Nature, A-H)

Referee #2 (Remarks to the Author):

Review is attached as PDF file.

Referee #3 (Remarks to the Author):

A. Summary of the key results

Barends and co-workers present an extensive study that demonstrates the vital importance of executing light-dependent, time-resolved serial femtosecond crystallography (TR-SFX) experiments in the single-photon regime. They used spectroscopy, crystallography, and computations to evaluate the photodissociation of CO from the myoglobin (Mb) heme center with high temporal and spatial resolutions, and from samples in solution as well as in microcrystal slurries. The authors support their experimental observations with quantum chemistry molecular dynamics analysis. They document that the illumination power has a significant impact on the photodissociation process and the refined atomic modes for the TR-SFX data sets. The analysis shown in Extended data Figure 1d indicates “the linear photoexcitation regime; it is ≤ 10 mJ/cm².” For some context and considering Extended Data Table2, this regime appears to be under an average of one photon per heme (18 mJ/cm²).

B. Originality and significance: if not novel, please include reference

Although it is not a photosensor nor a light-dependent system physiologically, the Mb-CO system is one of the best studied examples in biophysics, and thanks in part to these results, scientist now have data that can be more easily and directly compared across disciplines. The authors have applied similar strategies and samples at the LCLS for results published in Barends, et al. “Direct observation of ultrafast collective motions in CO myoglobin upon ligand dissociation.” *Science* 350, 445-50 (2015). However, those results were generated with about 5 photons per heme - well into the multiphoton regime. In contrast, the current work includes results generated with an average of between 0.2 – 5.8 photons per heme. The results potentially impact many other light-activated systems with time-resolved XFEL-based studies, which are briefly discussed most often by calling specific references and calling into question the validity of their conclusions if/when “overly high photoexcitation energies” were used.

C. Data & methodology: validity of approach, quality of data, quality of presentation

The authors exploited significant beamtime at SwissFEL and have done an excellent job detailing and documenting their experimental procedures. They have deposited about 45 atomic models with the PDB, many are listed as "processing complete, entry on hold until publication." The authors collected five TR-SFX data sets with a 10 ps delay time, each at a different illumination power from 6 mJ/cm² to 101 mJ/cm². They also collected time-series TR-SFX data at three powers (5, 23, and 101 mJ/cm²) with between 10 – 14 timepoint datasets (from 254 fs to 1373 fs) at each illumination power. Nearly all the datasets include at least 20,000 lattices (some many more) and the data typically extends to better than 1.4 Å resolution. Some of the text annotations in the figures will be too small to read (e.g. Figure 1d).

D. Appropriate use of statistics and treatment of uncertainties

Line 404 – 446: The authors state that, "structure factors were extrapolated to full occupancy using the linear extrapolation approximation (Ref 46,47)." However, there is not enough information given so that others working with the data could reproduce with certainty the authors' analysis in this study. The authors go on to state that, "the methods recently evaluated by de Zitter and coworkers (Ref 53) did not result in appreciable improvements. We therefore did not apply these methods." One of the strong benefits to the community from the approach of Zitter et al is that the extrapolation approximation analysis is well documented and can be easily reproduced by others. The current manuscript does not meet that standard and more information is required for this type of analysis.

Line 448 – 453: "To obtain error estimates for structural parameters such as bond lengths and ..."
The authors should indicate what the estimated coordinate errors are for these studies. This information should also be included in Extended Data Table 1. The plot shown in Figure 1c might also indicate error bars.

E. Conclusions: robustness, validity, reliability

The data was collected at SwissFEL with best practices and state of the art methods. The analysis is sound and generally useful to the community. However, the extrapolated maps generated by the authors may be difficult to reproduce by others (see comment above, Line 404-446). The authors are critical to several other published TR-SFX results. The current work appears to have benefitted from significant SwissFEL beamtime. It might be useful to suggest a minimum approach that others might consider when XFEL beamtime is scarce.

F. Suggested improvements: experiments, data for possible revision

Line 71: It is much better to spell out "sub-picosecond" and "nanosecond" rather than use abbreviation with a number before the ps or ns usage.

Line 97-99: "Our first power titrations..."

Please state that this is applied to samples in solution.

Line 358 "The protein crystals were introduced into the XFEL beam in a thin jet using a gas dynamic virtual (GDVN) nozzle injector (Ref 38)."

Please indicate the typical crystal density.

Line 383 "... 20,000 images..." and elsewhere as appropriate.

This should probably read "20,000 indexed lattices" since it is possible that some images may contain more than one lattice.

Lines 512 – "Figure 1..."

a) Difference electron density...

Please indicate that these are Fobs(light) - Fobs(dark) difference electron density maps and if they are extrapolated maps.

Please indicate the resolution of the maps in the legend.

The "Fluence" labels may be too small to read when published; please enlarge them.

Please annotate the image to label the electron density feature indicative of CO*. If not too busy, one could also label the feature for the Fe out of plane location. Please label His94 and His63 and Phe43 residues in the image since these are referred to in several other places in the manuscript.

b) Apparent occupancy of the CO* state...

Please be sure to define CO* in the legend as the photodissociated transient state.

c) The legend should probably read, "Iron-out-of-plane (OOP-subscript-Fe) distance..." The reader may also wonder what is the uncertainty in the distances shown in the plot; can error bars be added?

d) Ca-Ca-distance change matrices...

The text numbers along the two axes are too small to read. The axes should include labels such as "residue number"

The "Fluence" labels may be too small to read when published; please enlarge them to match the top.

Figure 2.

The yellow text on a white background is very low contrast and will be hard to read.

Line 587 "a. Power titration data"

Please indicate that these data were collected with a 10 ps delay time.

Extended Data Table 1 (line 588 -):

"No. images" should probably be the number of lattices.

Line640: "allows identification of the linear photoexcitation regime; it is ≤ 10 mJ/cm²."

Please add context to this statement. Considering Extended Data Table2: Laser and excitation parameters, this regime appears to be under an average of one photon per heme (18 mJ/cm²).

Line 713 "Extended Data Fig. 6" – Please add residue names to the right-hand illustration.

G. References: appropriate credit to previous work?

Line 65: It would also be good to add this reference: Chapman, H. N., Annu. Rev. Biochem. 2019. 88:35–58; "X-Ray Free-Electron Lasers for the Structure and Dynamics of Macromolecules" <https://doi.org/10.1146/annurev-biochem-013118-110744>

H. Clarity and context: lucidity of abstract/summary, appropriateness of abstract, introduction and conclusion

The manuscript is well written, the illustrations are appropriate and often clear; however, some of the annotations will be very difficult to read.

Line 30: "However, all ultra-fast TR-SFX studies to date have employed such high pump laser energies that several photons were nominally absorbed per chromophore (Refs 2 -14)."

Some readers may wonder about "all" in this statement. Indeed, the first reference called is Ref 2, (Barends, T.R. et al. Direct observation of ultrafast collective motions in CO myoglobin upon ligand dissociation. Science 350, 445-50 (2015) DOI: 10.1126/science.aac5492). Within Ref 2, these same senior authors state: "The laser energy was set to 5 μ J/pulse. This corresponds to \sim one photon/heme given the experimental parameters, the protein concentration in the crystals (51 mM), and an average crystal thickness of 6 μ m. (This value has a large variation given the plate-like shape of the crystals and the fact that we had no control over their orientation.) Thus, assuming a 90 μ m spot, the power density was 380 GW/cm²."

Considering the natural evolution of R&D projects, and e.g. Ref 16 (Brändén, G. & Neutze, R.

Advances and challenges in time-resolved macromolecular crystallography. *Science* 373, eaba0954 (2021), it appears that Barends et al (2015) may have used about five photons/heme for their prior Mb-CO photodissociation studies rather than one photon per heme. The current manuscript should probably very briefly address this explicitly in the abstract and/or introduction. Fortunately, this is addressed much more fully in Extended Table 2 and in several other places deeper in the manuscript and supplemental material.

Author Rebuttals to Initial Comments:

Report by Referee 1:

We thank the referee for his/her analysis of our work.

The only minor point I can make is that extended figures 7 and 8 do not have the b panels labelled and the discussion is a bit confusing.

We added the labels of the (a) and (b) panels.

Report by Referee 2:

We thank Richard Neutze for having invested so much time and many thoughts to review our manuscript. Before addressing the specific issues raised, we would like to emphasize two central points where we seem to disagree, possibly because of misunderstanding.

1. Time-resolved experiments are performed to characterize reaction intermediates with the goal to derive a chemical mechanism. This means that the reaction coordinate, the path that a reaction takes, matters, not just the outcome, be it a certain intermediate or the final product. In contrast to spectroscopy experiments that continuously follow the temporal evolution of the reaction, observing build-up and decay of intermediates, due to sample and beamtime limitations TR-SFX experiments in most cases only probe the evolving reaction at time points where the peak occupancy of intermediates is high.

Akin to the well-known saying that all roads lead to Rome, it seems that some (many?) reaction intermediates may form independently of the reaction conditions and the reaction pathway taken. However, even in cases where this is true it does not help with the characterization of the reaction coordinate and thus the reaction mechanism. Various ultrafast time-resolved spectroscopic investigations on different heme proteins and rhodopsins have indicated that the rapid electronic changes induce an impulsive nuclear response composed of those vibrational motions that are coupled to the electronic state changes (see Champion Science 2005, <https://www.science.org/doi/10.1126/science.1120280>). According to the mode-coupling model (Miller, Acc. Chem. Res, 1994, <https://pubs.acs.org/doi/abs/10.1021/ar00041a005>) the fast modes couple to slower protein modes, driving the structural changes occurring during the reaction. This means that the nature of the electronic transitions matters, since they drive specific nuclear vibrations; moreover, excess absorbed energy can affect magnitude and dynamics of conformational changes. Thus, taken together, when using TR-SFX to deduce a mechanism it is necessary but not sufficient to determine structures of reaction intermediates. The characterization of the dynamics of the system is fundamentally important too, in particular because it allows direct comparison with spectroscopic insight.

The first reaction intermediate of the photodissociation of Mb.CO studied in our manuscript is the so-called B-state (denoted CO* in this manuscript) where the dissociated CO is oriented atop the heme plane.

Irrespective of experimental approach (cryotrapping, Laue, SFX) and illumination condition, the B-state structures resemble each other (with the exception of Teng et al *Nat. Struct. Biol.* 1, 701–705 (1994), see Schlichting & Chu *Curr Opin Struct Biol* 2000), as pointed out in the Branden & Neutze Science review. However, none of the structures provided insight into the reaction mechanism itself. First insights were obtained in our previous TR-SFX study using multiphoton excitation (Barends et al Science, 2015) that showed correlated oscillations of protein residues with a frequency reminiscent of the spectroscopically determined heme doming frequency. This observation is in line with Miller's mode-coupling model (Miller, Acc. Chem. Res, 1994, <https://pubs.acs.org/doi/abs/10.1021/ar00041a005>), suggesting how the energy absorbed by the chromophore is transmitted to the protein matrix and transformed into protein structural changes. Our current study, performed under single photon excitation conditions, shows an unexpected apparent increase of the photolyzed CO occupancy with time. We correlate this with a

predicted damping of a fast, computationally observed CO oscillation. The underlying quantum chemistry analysis links electronic and nuclear effects and shows how the excited state properties affect the Fe-CO bond dissociation. Although we cannot resolve the predicted CO oscillation yet – beamtime proposals have been submitted at SwissFEL and LCLS – it is an important step towards understanding the mechanism of a fundamental organometallic reaction. This is the scientific finding in our manuscript. It is intimately linked to a technical aspect, which is that in this case the choice of photoexcitation conditions is crucial.

The referee acknowledges that the “structural differences in the myoglobin:carbon monoxide system are very subtle but may affect the sub-picosecond kinetics.” Indeed, this is our point. The dynamics and magnitude of the structural changes differ with pump laser fluence, in line with quantum chemistry calculations that show that the CO photodissociation mechanism changes upon multiphoton excitation.

2. A mechanistically clean comparison of the results of TR-SFX and spectroscopy pump probe experiments requires that both approaches use similar excitation conditions. In case of strong discrepancies, for example single vs multiphoton excitation, different reaction pathways may be taken, radicals may form, or changes in magnitude and dynamics of conformational changes may occur (all of this occurs in carboxymyoglobin). In other words, multiphoton excitation can affect the reaction coordinate and mechanism, respectively; it simply differs from the single photon excitation situation. To relate this difference to damage or artifacts in the electron density is misleading, since damage has a very different connotation. In the context of crystallography, damage is associated with radiation damage. In that case, damage frequently results in molecules breaking or falling apart (decarboxylations etc) which occurs extremely rarely upon photoexcitation using visible light.

Whether or not radiation damage occurs is established by comparing electron densities (or structures) derived from data collected with different absorbed dose. This would correspond to comparing DED maps obtained with different pump laser fluences ($F_{\text{obs}}(\text{fluence1}) - F_{\text{obs}}(\text{fluence2})$). However, this comparison, and the juxtaposition of $F_{\text{obs}}(\text{light}) - F_{\text{obs}}(\text{dark})$ DED maps as a function of laser fluence without further analysis (Fig. 1a in the current manuscript, Fig. S4 Yun et al PNAS 2021, Fig. 2 Claesson et al Elife, 2020, ...) is **not** good enough to analyze the effect of excitation conditions on the reaction, since two effects may contribute to the differences: increased signal to noise due to increasing occupancy with fluence and possibly actual fluence-related conformational differences. Structural refinement is required to distinguish between these contributions. This has not been done so far, despite the fact that such data is available (see references above). What has not even been attempted so far is to collect an entire time-resolved series at different pump power densities to check whether the structural outcome really is the same. Our current myoglobin study is the first one to do this.

Influence of pump laser fluence on ultrafast structural changes in myoglobin, Barends et al.,

This article describes important work addressing whether or not the power density of an incoming laser pulse that is used to photoactivate a light-sensitive biological sample has a measurable effect on structural changes observed using time-resolved X-ray crystallography. Care has been taken in the experimental design and the quality of the crystallographic data are excellent. The myoglobin:CO system is a good choice for such studies since the quantum yield for photo-dissociation is high and small microcrystals diffract very well when exposed to an X-ray free electron laser (XFEL) beam.

There is great interest in this work since there is a discrepancy between the photoexcitation conditions used historically for time-resolved X-ray diffraction studies, and those used in time-resolved spectroscopy studies. Specifically, in order to achieve the maximum photo-excitation yield

within crystals, all time-resolved X-ray diffraction studies reported to date (1) have used an integrated pump laser fluence that is much (typically by one to two orders of magnitude) higher than would normally be judged to be acceptable by the time-resolved spectroscopy community. This risks the interpretation of structural results from time-resolved X-ray diffraction studies being judged to be oversold or potentially biologically irrelevant. Conversely, in the quarter-century since the first demonstration of light-induced conformational changes in crystals of myoglobin using time-resolved Laue diffraction (2), no study has convincingly shown multiphoton artefacts in the experimental difference Fourier electron density maps.

Please see our general comments above concerning (i) characterizing structures of reaction intermediates vs additionally analyzing the reaction dynamics and (ii) the use of the term “damage”. Along the same line, without referring to “damage” strong mechanistic objections have been published (Hughes, *eLife*, 2021) regarding the TR-SFX results on photoexcited phytochrome (Claesson, *eLife*, 2021). Importantly, so far no quantitative structural analysis of single vs multiphoton excitation has been published and it is thus not surprising that little crystallographic insight exists.

Thus, on the face of it, there seems to be a serious problem that the field has not addressed, yet no clear evidence of the problem is observed in the crystallographic data to date. The authors summarize this issue as:

“As multiphoton absorption may force the protein response into nonphysiological pathways, it is of great concern whether this experimental approach allows valid inferences to be drawn vis-à-vis biologically relevant single-photon-induced reactions”

By performing careful power titration studies using a light sensitive model system, Barends *et al.* take the only viable approach to address these concerns. Specifically, time-resolved serial femtosecond crystallography (TR-SFX) is used to follow the photodissociation of carbon monoxide from the active-site of myoglobin with sub-picosecond time resolution, and these experiments are repeated with different pump laser fluence. The difference Fourier electron density maps from this study (Fig. 1A, pasted below) are very convincing and show a clear trend of improved signal-to-noise with increased pump laser fluence. However, when I look at this figure, I do not see any indications of laser induced damage or structurally distinct multi-photon pathways at higher laser fluence, and no such effects are suggested by the authors to be visible in these difference Fourier electron density maps.

The referee makes an important point. Looking at difference electron densities collected for a single time point is necessary but not sufficient to judge whether the pump laser fluence changes the reaction coordinate. As already mentioned above, increasing the laser fluence can result in two effects, an increase of signal/noise and fluence-related structural changes. Thus, the very least that needs to be done is to refine the structures and to plot the magnitude of the light-induced changes (for example Fe-out-of-plane, photolyzed CO occupancy) as a function of laser energy. Unfortunately, so far the community used a mere visual inspection of the DED maps to decide on the pump laser parameters.

Please see also our point above relating to “damage”.

“structurally distinct multi-photon pathways at higher laser fluence” are apparent in the DED shown in Ext. Data Fig. 6

What is clear is that the overall quality of the map improves as the pump laser fluence is increased, which appears to support the use of relatively high pump laser fluence in time-resolved diffraction studies.

See above.

Barends et al. therefore analyse the results from structural refinement against extrapolated data to search for pump-laser fluence dependent artefacts in the structural interpretation. This approach is less direct than inspecting the difference Fourier electron density maps, since many assumptions are made during structural modelling.

It is true that structural modeling is more complex than a straightforward visual inspection; in particular, the determination of the occupancy of the intermediate has to be done very carefully. While positive and negative difference electron density peaks enable picturing structural differences (from here to there), the insight obtained is qualitative at best. Detailed structural analysis requires structural modeling and refinement.

Moreover, the article currently lacks sufficient internal controls to demonstrate unambiguously structural differences as the pump laser fluence is increased. Thus, whereas I am favourably inclined towards this work, I think for such an important study it is essential that the proper internal controls are performed.

We fully agree.

Moreover, although I appreciate that the authors are looking carefully for pump-laser fluence dependent differences in structural interpretation, the overriding message of their structural analysis is that, where such differences exist, they are subtle: for example, the protein appears to respond a bit faster in the sub-picosecond time-domain at high pump laser fluence.

Conversely, there is an overriding agreement between the nature of conformational changes at all pump-laser fluence values and this is a major result of this work. Or (to rephrase the authors' own wording) no compelling evidence is presented to demonstrate that multiphoton absorption forces the protein response into nonphysiological pathways.

Here we disagree very strongly. The referee is focusing on the structure of the B-state intermediate, which is indeed similar for three photoexcitation data sets. An important point neglected by the referee is, is whether "overriding agreement" of reaction intermediate structures is good enough. The overriding agreement of the structures is that the iron moved out of the plane and the dissociated CO is located parallel atop the heme plane. So far so good. However, reaction mechanisms in particular of metalloproteins are typically investigated by TR-XANES, EXAFS, RIXS,... X-ray spectroscopies that probe the coordination of the metal in very fine detail, at very high local resolution. Different proposed reaction mechanisms are often distinguished by the specific bond length(s), differing on the order of 0.1 Å, with photosystem II being a prominent example.

Importantly, it is not only the structural details that matter when it comes to analyzing possible reaction mechanisms, but also the dynamics of the structural changes. They differ fundamentally for the three

photoexcitation fluences. In particular, apparent CO-dissociation differs quite dramatically, according to the quantum chemical calculations due to different electronic properties of the single and double excited states. The excess energy dumped into the two-photon excited protein results in significantly faster structural changes (dynamics) of the protein matrix, in line with previously published spectroscopic analysis.

I am not surprised by this since, had nonphysiological pathways been the dominant population in crystals, then the contradictions associated with structural results would have been clear decades ago.

The referee misses the point: First, all previous time-resolved experiments on MbCO were performed using multiphoton excitation. It is thus not unexpected that they agree with each other (see Branden and Neutze Science review). No crystallographic study has been published using single-photon excitation. Second, except for our TR-SFX experiment published in Science, previous crystallographic studies on photodissociated MbCO were limited to structure determination of the CO* (B-state) intermediate. Our study does not focus on the long-known CO* structure but instead on the pathway towards its formation. This is a very important point that the referee seems to have missed, hence the introductory section at the beginning.

For the sake of clarity: my recommendation to the editor handling this work is therefore that the authors be invited to revise and resubmit the work to *Nature*, but with significant changes in their analysis and more balance in the conclusions presented.

Difference Fourier electron density maps:

For almost fifty years, the gold standard when looking for structural differences in X-ray crystallography data has been the difference Fourier electron density map. This is because Henderson and Moffat showed that this approach is more sensitive than any other structural analysis tool (3). In a recent email communication with one of the leaders of the field of time-resolved X-ray crystallography, I received the following comment:

“If significant in our fs experiments, damage MUST be evident in DED [difference electron density] maps corrupted by two-photon absorption. If present but not visible in these maps, it's irrelevant. I don't know of any features in DED maps in any system that are claimed to arise from damage. do you? If even one feature existed, then adding a pump power titration to the experiment should produce variation in the magnitude of the damage feature(s).”

Again, “damage” is not the appropriate term, see section at the beginning. The expert correctly points out that a pump power titration should produce variation in the magnitude of the “damage” feature(s), which is exactly what we show in our manuscript.

I am sure that the authors will have looked closely for evidence in their difference Fourier electron density maps of multi-photon induced damage, or multi-photon induced pathways that are distinguishable from the primary photo-dissociation reaction.

Indeed, this is the entire point of the current manuscript. We describe plenty of evidence. The most obvious difference is the dependence of the CO* occupancy with pump-probe delay times $\Delta t < 1\text{ps}$ (Extended Data Fig. 6)

Some examples of possible multiphoton artefacts have been suggested by the authors, writing in in reference (4):

“Multiphoton excitation can amplify nuclear motion (see the supplement in ref. 3), lead to additional radical intermediates^{5,6} and open nonproductive higher excited state relaxation channels, decreasing the single-photon reaction yield⁶⁻⁸.”

Or in reference (5) they write (referring specifically to retinal proteins):

“Quantum chemistry and ultrafast spectroscopy were used to identify a sequential two-photon absorption process, leading to excitation of a tryptophan residue flanking the retinal chromophore, as a first manifestation of multiphoton effects.”

In the current study, Barends *et al.* carefully performed TR-SRX studies with the pump-laser fluence ranging from 2.4 mJ/cm² to 101 mJ/cm², the latter more than an order of magnitude above what the authors indicate is a threshold for the onset of multiphoton effects. Yet all they write regarding the difference Fourier electron density maps is:

“Inspection of $F_{obs}^{light} - F_{obs}^{dark}$ difference electron density maps shows a clear change of the magnitude of the peaks associated with bound and photolyzed CO, respectively, and the iron. At higher laser fluence, changes are also apparent in the protein and the porphyrin ring (Fig. 1a). Considering only the difference density as in previous TR-SFX studies^{7,9,16,22,23}, a laser fluence of 101 mJ/cm² appears preferable.”

The referee misses the point entirely. He is focusing on the power titration data which were performed to establish the linear photoexcitation regime (defined as a linear increase of signal (in this case occupancy of CO* with fluence).

The authors do not claim that the movements associated with the porphyrin ring, seen at higher pump-laser fluence, indicate multiphoton artefacts. Indeed, my understanding is that the porphyrin ring movements are universally accepted to be an important structural response to the photo-dissociation of CO from the active-site haem of myoglobin, and their presence in the difference Fourier electron density maps at higher pump laser fluence simply reflects the improved signal-to-noise when there is higher photo-dissociated state occupancy. In the supporting information the authors comment on the difference Fourier electron density maps and, not unreasonably, write:

“The objective, however, must not be to simply increase peak heights until usable signal results. Valid characterization of biologically relevant single photon-induced reactions can only be made in the regime of linearly increasing signal with pump laser energy.”

In summary, Barends *et al.* perform TR-SFX studies with the pump-laser fluence ranging from 2.4 mJ/cm² to 101 mJ/cm². The latter condition the authors argue is dominated by multiphoton effects, yet no evidence of off-pathway reactions or multiphoton induced damage are evident in their difference Fourier electron density maps. It is therefore a very strong conclusion that multi-photon absorption effects cannot be identified directly from the difference Fourier electron density maps, and this should be clearly stated in the abstract and in the conclusions of the paper.

This is not correct as stated. It very much depends on the time delay. While the difference density maps calculated for the 10 ps time delay appear relatively similar, this is not the case for the sub ps time delays. In case of the low and medium fluence data the magnitude of the difference electron peak associated with the photodissociated CO* increases with time (see Extended Data Fig. 6), in case of the high fluence case it does not. This applies for the high fluence data reported here as well as for the earlier Barends *et al* 2015 data.

Long-distance view of the difference Fourier maps

In order for the reader to judge the level of noise in the difference Fourier electron density maps, a long-distance overview of these maps should be shown. A representation similar to that used in Figure 1A of Barends *et al.* (2015) (6) would be fine, and these should be presented for every pump laser fluence. This would allow the reader to judge the signal to noise in their data.

Here it is for the power titration data. The maps are contoured at ± 3 sigma. The corresponding maps for the time-resolved data series are shown in Extended Data Fig. 5.

Similarly, no difference Fourier electron density map for the lowest pump-laser fluence value of 2.4 mJ/cm² is shown in the paper. Instead the authors write:

“The 2.4 mJ/cm² data did not yield interpretable light-induced signal and will not be discussed further.”

But this is actually a very important result, since the occupancy would be expected to be about 10 % at 2.4 mJ/cm², given the estimated 20 % occupancy of the 5 mJ/cm² time-series (Table 1). As such, both a close-up and long-distance overview of a representative example from the lowest pump-laser fluence difference Fourier electron density maps should be shown in a revised article.

We thank the referee for pointing this out; this has been added as Extended Data Fig. 5. We also reanalyzed to 2.4 mJ/cm² data. As detailed in Supplementary Note 1.4 the structures cannot be refined using extrapolated structure factors. Ensemble refinement is borderline due to low signal to noise. Our simulations strongly suggest that the 2.4 mJ/cm² data could be refined if the signal increases by a factor of two, which is entirely possible experimentally (required sample, beamtime). We have added this to the manuscript.

Crystallographic occupancy

Since the authors do not directly observe artefacts that can be assigned to multiphoton absorption processes in their difference Fourier electron density maps, they examine the results of structural refinement in order to further investigate the influence of multiphoton effects in their crystallographic data. This is a reasonable strategy to pursue, but it is also fraught with potential pitfalls since the conclusions drawn depend upon the assumptions made during structural refinement and in the analysis of these structures.

The first statement is not correct since, it implies that we would not have refined the structures if we had seen large differences in the power titration data (we do see differences between the

subpicosecond difference electron density maps of the 5, 23 and 101 mJ/cm² data). In order to compare structural differences and their evolution, one needs to have structures. Difference densities are simply not accurate enough; they only allow deducing that something switched from here to there. It is for this reason that techniques such as structure factor extrapolation were developed, which allow building models of the triggered state; difference maps do not allow this.

The first assumption to consider is how the crystallographic occupancy of the photo-dissociated state is determined. This is absolutely central to many of the conclusions drawn, since then the coordinate displacements will be overestimated during structural refinement if the occupancy is underestimated and *vice-versa*. What is concerning is that the results do not seem to be consistent. Specifically, in Fig. 1b (below, left), the occupancy estimated from the power-titration shows a maximum occupancy of 40 % for a power density of 101 mJ/cm². In Fig. 2a, the exact same formula (illustrated in the schematic in the RHS below) appears to give a maximum occupancy of approximately 75 % for the time-resolved diffraction data using a power density of 101 mJ/cm². Obviously, the extent of coordinate displacements resulting from structural refinement will differ if the crystallographic occupancy is assumed to be 40 % or 75 %. Similar (albeit not to the same extent) discrepancies are apparent for occupancy estimates using power densities of 23 mJ/cm² and 5 mJ/cm².

Concerning the issue of crystallographic occupancy, the authors write

“Importantly, we found that the best apparent occupancy to be used for extrapolation (that is, the occupancy that results in maps that exclusively show the photolyzed state) differs from those found from difference electron density maps and is sensitive to resolution limits, weighting schemes etc. and must be determined a new for each case. This was done by increasing the assumed occupancy until dark state features became apparent in the extrapolated electron density maps⁴⁶⁻⁴⁸, i.e. where density for unphotolyzed (i.e. heme-bound) CO became visible.”

Thus, a different method was used to estimate the crystallographic occupancy for structural refinement than that used to calculate the data shown in Figures 1b and 2a, which may explain some of the inconsistencies, but which is correct? Whereas crystallographic occupancies are not given in Extended Data Table 1a (I suspect this is a typo rather than a scientific error), in Table 1b this is given as 20 % for 5 mJ/cm²; in Table 1c this is given as 30 % for 23 mJ/cm²; in Table 1d this is given as 42 % for 101 mJ/cm².

Thank you very much for having caught this. Stimulated by the comments and criticism we have reanalyzed the data from scratch, comparing different methods for occupancy determination and structural refinement. The various approaches were benchmarked using simulated, partially photolyzed data sets. The approaches and results are now described in detail in a new Supplementary Note 1, containing a Methods and Analysis section. Many of the points mentioned further below are

addressed there too.

An important finding is that for TR-SFX data it is unexpectedly difficult to determine the occupancy. Different methods give different absolute values (as also mentioned by the referee above). However, the trends are the same, such as the form of the curve shown in fig.1 b (see also previous page). For all methods used, the occupancy increases with fluence and the slope changes from steep to modest. Depending on the method chosen, the absolute value of the occupancy at 101 mJ/cm² is between ~0.4 and ~0.8 (see Supplementary Note 1).

These values should be compared to Barends et al., 2015 (6) in which the crystallographic occupancy for the photo-dissociated species was given as 100 %. I have checked the 2Fo-Fc electron density maps associated with Barends et al. (pdb ID 5CNE vs. 5CMV) and I am very satisfied that, for those crystallographic data, the crystallographic occupancy of the photo-dissociated state was indeed very close to 100 %, as was the crystallographic occupancy of CO in the CO-bound dark-state also very close to 100 %.

Barends et al. use bootstrapping resampling (a technical name for a method of randomly selecting images from serial crystallography data-set (8, 9)) to estimate coordinate errors. The same approach could be used to estimate uncertainty in the crystallographic occupancy. In my lab, we have tested bootstrapping resampling of serial crystallography data to estimate occupancy uncertainty using the program Xtrapol8 and the results seem quite reasonable. Bootstrapping resampling should also allow uncertainty in the crystallographic occupancy to be estimated, by comparing the bootstrapping resampled photo-activated data-sets (100 have been selected) with bootstrapping resampled dark data-sets (100 have been selected). All of the plots indicating crystallographic occupancy (eg. Figs. 1b and 2a) should then be redrawn to show uncertainty bars for the crystallography occupancy. The crystallographic occupancy for the 2.4 mJ/cm² data should also be estimated using the same approach, and the predicted occupancy and uncertainty in the occupancy of the 2.4 mJ/cm² data should also be shown in all of these occupancy plots. I would expect the analysis of the 2.4 mJ/cm² data sets to give larger uncertainties in the occupancy estimates since, as noted above, the difference Fourier electron map was not convincing with those data. As an additional check, the dark- data stream-files can be split into two halves, and the occupancy uncertainty can be estimated for the “zero pump-laser fluence” data-set (which is shown as being exactly 0 in the plots pasted in above, but I assume that this is a postulate rather than a result recovered from a parallel analysis of the dark-state crystallographic data). The impact of uncertainty in the crystallographic occupancy on the uncertainty in the crystallographic coordinates should then be evaluated to the best of the author’s ability, and this factor should be explicitly acknowledged as one of the largest sources of uncertainty in their structural analysis.

As mentioned above, we analyzed the issue of occupancy determination in great detail and described our findings and reasoning for using a certain consistent approach for all data in the new Supplementary Note 1. Briefly, we now use two methods for the occupancy determination: one based on peaks in Fo-Fc omit maps, and one based on refinements of ensembles of light- and dark-state models. The former method makes very few assumptions, relying on peak heights in maps phased using a model containing no CO molecule. The latter method refines a range of light/dark ensembles with different light-state occupancies and optimizes the difference density on the dark-state CO position. This ensemble refinement method was benchmarked against simulated data, using simulated error levels comparable to those encountered in the actual observed data. Moreover, the two methods give very consistent results.

The ensemble refinement method was used to determine the occupancies that were used for the determination of photolyzed structures (which was also done using ensemble refinement). We now use bootstrapping to obtain error bars on these occupancies (in Figure 1 and Extended Data Figure 7a).

Given its poorer performance in the simulations, we decided to no longer use structure factor extrapolation for either occupancy determination or structure determination. We also did not use XtraPol8 for the final analysis of all data sets as the ensemble refinement method we decided on for data analysis is not (yet) available in XtraPol8.

Thus, an important finding of our study (triggered by this review) is that it is exceedingly difficult to determine the absolute value of the occupancy in TR-SFX data.

This also means that absolute numbers (rotation by x degree, bond length of y Å) can have very large error bars. Fortunately, most of our findings depend on comparing relative changes with time within a certain data set. This allows the derivation of time constants which should not depend strongly on the absolute occupancy value.

While I appreciate that error bars associated with crystallographic occupancy are not usually presented in the scientific literature, I am not aware of any other paper in the field of time-resolved X-ray crystallography for which such far-reaching conclusions hinge upon an accurate estimate of the crystallographic occupancy.

As mentioned above, we calculated these error bars, see Supplementary Note 1. As pointed out there in detail, different methods give different absolute values for the occupancy. Importantly, however, the features of the occupancies (and of the derived maps/structures) are the same. In particular, in the 10 ps power titration data the occupancy of CO* increases and then saturates, and the occupancy of CO* increases with sub-picosecond pump probe delays for the low and medium fluence excitation data, but increases very rapidly (“leaps”) for the high fluence data. Similarly, for the three time-resolved data series the apparent occupancy of CO* increases with the same time dependence irrespective of the method used to determine the occupancy and thus of the final occupancy value.

Refinement against extrapolated data

The authors use structural refinement against extrapolated data. There are some variations in how this is implemented by different groups, and there is also the option to use partial occupancy refinement rather than refinement against extrapolated data (1). I do hold some reservations about the wisdom of structural refinement against extrapolated crystallographic data, largely because the excited state phases cannot be known when extrapolating the data (eg. see Figure 1 of reference (10));

and errors in the crystallographic data are exaggerated by the extrapolation procedure, especially when working with low crystallographic occupancy. However, the Heidelberg group will be completely familiar with all of these concerns, and they have brought very high crystallographic standards to the field of serial crystallography over the last thirteen years. I am therefore going to accept that it is valid to draw structural comparisons using structural refinement against extrapolated data as long as everything is done in a self-consistent manner, as is the case in this work.

We entirely agree with the caveats of using extrapolated structure factors. As already mentioned, we reanalyzed the data and compared the results obtained by refinement using either extrapolated structure factors or ensemble refinement, benchmarking with simulated data as described in the new Supplementary Note 1. In case of our high-resolution myoglobin data, ensemble refinement performs better in simulations in terms of being able to retrieve the “true” light-state structure, in particular for the low occupancy data. Therefore, all structures described in the main text were obtained with ensemble refinement.

Internal distance matrix analysis

I really like that the authors use an internal distance matrix analysis to look for systematic differences in the results of structural refinement as the pump laser fluence is varied. An advantage of this tool is that internal distance difference matrices can be calculated without first aligning dark and photo-activated structures upon each other. When reflecting upon the results from this analysis the authors first write:

“Although difference-distance matrix plots do not seem to show significant structural differences as a function laser fluence (Fig. 1d, Extended Data Fig. 3a)...”

Thus, as with my comments above concerning the author’s analysis of the difference Fourier electron density maps, the authors indicate that it is very difficult to observe structural differences as the pump laser fluence is increased when considering internal distance changes on C α -atoms throughout the protein. Again, I consider this to be a strong and important conclusion of this work.

See general comments at the top. As mentioned before, the 10 ps data point corresponds to the long-lived B-state intermediate. While the 10 ps time delay structures determined from the different photoexcitation data sets differ somewhat the major differences are observed in the reaction coordinate and dynamics for the sub 1ps data.

Displacements of C α -atoms relative to the haem

The above quote continues:

“... the analysis of the displacements of C α atoms from the porphyrin nitrogen atoms indicates that such differences are indeed present (Extended Data Fig. 3b). Hence the influence of multiphoton excitation on structural changes it is not always immediately obvious and may demand very careful analysis.”

Let's therefore focus on the displacements of C α atoms relative to the porphyrin nitrogen atoms. The important figure here is Figure 3a (left). What is striking about this figure is that the nature of the displacements of the backbone C, N and C α atoms of the protein relative to the haem is conserved as the pump laser power is increased. Moreover, it is not surprising that the 5 mJ/cm² data show larger fluctuations than the other data sets, since structural refinement against extrapolated data exaggerates crystallographic errors when the occupancy is low

The referee misses the point. We do not display these graphs to argue that there are differences in structure, but rather that there are difference in dynamics, see also the beginning of this discussion.

A Pearson correlation function provides a score for the agreement between two curves which is independent of the amplitude of these curves. This is useful in this context because it largely side-steps my concerns regarding the uncertainty in the crystallographic occupancy. I therefore suggest that the authors utilize the Pearson correlation coefficient if they wish to establish that the shape of the curves plotted above depend upon the pump laser fluence in a manner that does not depend critically upon an accurate estimate of the crystallographic occupancy.

As mentioned above, we do not want to show differences in the end points, but differences in the dynamics with which those end points are reached. These are clearly different, as can be seen in the plots for individual residues.

Specifically, from the bootstrapping resampling procedure utilized by the authors, 100 "light" structures can be calculated for each photo-dissociated data-set, and 100 "dark" structures can also be calculated from the dark-state data. These can then be compared against each other, from which 100 curves similar to those shown above (Figure 3a) can be calculated (or 10,000 if all 100 cross-comparisons are made, but let's assume 100 curves are sufficient for this purpose). 9,900 Pearson correlation scores can then be calculated by comparing each of this set of 100 curves against the remaining 99 curves within the same set. From this set, a mean and standard deviation of the Pearson correlation scores can then be extracted. I would expect that the mean Pearson correlation coefficient would be quite high, and its standard deviation quite low, for the high pump-laser fluence data-sets since the signal-to-noise is very good. I would also expect that the mean Pearson correlation coefficient would become lower as the pump-laser fluence decreases, and its standard deviation should increase. These values would thus provide a measure of the internal consistency of the observed structural differences for each pump-laser fluence, given the bootstrapping resampling

procedure advocated by the authors.

See above

With this tool in place, the same procedure can then be used to compare the 100 displacement curves at one pump-laser fluence against 100 displacement curves using another pump-laser fluence. This will give 10,000 Pearson correlation coefficients from which a mean and standard deviation can also be calculated. If, for example, the structural changes observed at 5 mJ/cm² were statistically different to those observed at 101 mJ/cm², then I would expect the mean Pearson correlation coefficient between these data-sets to be much lower (ie. by a standard deviation or more) than that recovered from their respective internal comparisons. Conversely, if the internal comparison for the 5 mJ/cm² data-set consistently gave the lowest Pearson correlation coefficients, then I cannot see how it can be claimed that the structural changes associated with those data are statistically distinct from data recorded at higher pump-laser fluence.

See above

I hope that the authors consider this a constructive suggestion that may be useful. Unfortunately, I have not had the chance to benchmark this suggestion against any data in my own laboratory and there may be pitfalls that arise that I have not foreseen.

We absolutely do consider this a constructive suggestion, and we thank the referee as we hope to be able to apply this somewhere else, but the point of these graphs is not to show differences in structure, but differences in dynamics.

Haem iron displacement

Another very important conclusion from this study is that the iron out-of-plane displacement appears to have a pump-laser fluence dependence. This is shown in Figure 2b, which I paste below.

This graph (LHS panel above) indicates that the iron out of plane displacement becomes significantly larger as the pump laser fluence increases from 5 mJ/cm² (20 % occupancy) to 23 mJ/cm² (30 % occupancy) to 101 mJ/cm² (42 % occupancy). On the face of it, this seems like a pretty convincing result. However, as noted above, there is uncertainty in estimating the crystallographic occupancy of the photo-dissociated state. If the crystallographic occupancy has been underestimated for the high pump-laser fluence data-sets, then the coordinate displacements will be exaggerated during structural refinement in order to compensate.

Unfortunately, I find it impossible to reconcile the results of structural refinement above concerning the iron out of plane displacement with the structural results previously reported by the same authors (6). In their earlier publication, structural refinement was performed with 100 % occupancy, and the iron out of plane displacement approached 0.3 Å (circles in the RHS panel above), in perfect

agreement with the low-fluence studies in this work, and (as argued in reference (6)) in agreement with a lot of other biophysical studies. I have looked at the electron density maps available from reference (6) and I am convinced that the crystallographic occupancy of the photo-dissociated state was close to 100 %. Thus, the only way I can reconcile the current results with the earlier publication is to accept that, in the current work, the authors have underestimated the crystallographic occupancy of the higher pump-laser fluence data. This conclusion also seems to be suggested by Figure 2a (pasted above), which indicates a crystallographic occupancy of 75 % for the 101 mJ/cm² study, yet an occupancy of 42 % was used during structural refinement (Table 1d).

This effect largely disappeared with the reanalysis of the data (see Supplementary Methods). For the 10 ps time delay data the iron out of plane distance is 0.3 Å for all fluences, including the low fluence data. In contrast, for the time-resolved data the iron out of plane is smaller for the 5 mJ/cm² data than for the other two fluences. Whether this is real or not, we do not know. It would be consistent with ultrafast X-ray transient absorption spectroscopy data (Shelby et al, PNAS 2021, <https://doi.org/10.1073/pnas.2018966118>) showing a 0.9 ps time constant.

Conversely, because of signal-to-noise issues, as the crystallographic occupancy decreases it becomes increasingly difficult for structural refinement to accurately model conformational changes. This is already suggested in the author's statement:

"The 2.4 mJ/cm² data did not yield interpretable light-induced signal and will not be discussed further."

Under those conditions, the crystallographic data should have an occupancy of approximately 10 %, but apparently the difference signal is completely lost (data were not shown by the authors). My experience is that structural refinement against noisy, low-occupancy crystallographic data tends to yield lower amplitude protein motions, and this cannot be completely compensated for by lowering the estimated crystallographic occupancy since that increases the overall sensitivity to noise in the data. Thus, while I am very confident of the excellent crystallographic standards of the Heidelberg group, I am not completely convinced that the lower amplitude of the motions associated with the lower occupancy data-sets is a real conclusion, and may instead reflect real limitations associated with structural refinement against data with low signal-to-noise.

I could go on to examine other examples given in the manuscript (eg. Extended Data Fig. 5), but I think that the point is made. If the crystallographic occupancy has been underestimated for the high-fluence data-sets, and if the coordinate displacements are slightly underestimated for the low-fluence data-sets because of difficulties in accurately modelling structural changes against low signal-to-noise crystallographic data, then all of the author's functional conclusions regarding the relative amplitudes of certain motions evaporate.

The referee is absolutely right, and we are very grateful to the referee for pointing this out. In fact, the referee's remarks are what sparked our investigation into different methods for photolyzed structure retrieval/occupancy determination. However, given our simulation results and the consistency between the two methods we now use for occupancy determination, we now have high confidence in the methods we used for both occupancy determination and photolyzed structure retrieval. Indeed, whereas we previously cut our data at 1.6 Å resolution because of the errors introduced by structure factor extrapolation, we now use them to 1.4 Å resolution, as we no longer perform extrapolation but use the data directly. Moreover, all data were treated in exactly the same way, using automated scripts to remove any human factors. This also ensures that trends will be conserved even if absolute values are not.

This is described in great detail in the new Supplementary Note 1. It also contains a section on the analysis of the 2.4 mJ/cm² data.

Coordinate errors:

Irrespective of the discussion above concerning crystallographic occupancy, it is important to accurately represent coordinate errors in this work. In the case of 100 % occupancy, a rough estimate for coordinate errors is given by (<https://legacy.ccp4.ac.uk/newsletters/newsletter33/murshudov.html>):

$$\sigma_{free}^2 = 0.65 \cdot \frac{n_a}{n_o} \cdot R_{free}^2 \cdot d_{min}^2 \cdot C^{3/2}$$

where $\sqrt{\sigma_{free}^2}$ estimates of the overall coordinate error in the structure; where C is the completeness, R_{free} is obvious, d_{min} is the maximum resolution, N_a is the number of atoms included in refinement, N_o is the number of independent observations. I did not find N_o in the table of crystallographic data, but neither am I sure how the very high multiplicity of serial crystallography data should impact on these estimates. A naïve comparison with the examples given in the above homepage suggest coordinate uncertainty of the order 0.05 Å for the (100 % occupancy) resting conformation.

In the current manuscript, Barends *et al.* use bootstrapping resampling to calculate coordinate errors, as they have previously described (8, 9). I think that this is a good method and I accept the logic applied in this approach. Looking, for example, at the central panel of Extended Data Figure 5d (left), the coordinate uncertainties indicated are similar to the estimate above as approximately ± 0.05 Å for the photoexcited species (23 mJ/cm²).

The figure, however, suggests that the coordinate errors of the dark structure (shaded area) are much lower (approximately ± 0.01 Å) than for the photo-dissociated data and this should be explained.

The referee is absolutely right; however, these expected coordinate errors are average values for the entire structure, including both well-defined atoms such as the electron-rich, fixed iron atom, as well as poorly defined side chains on the surface. The position of the heme iron, and thus its distance to any other atom may be expected to be determined far more precisely. We investigated this, looking at the distribution of individual errors as determined by bootstrapping and find that indeed, the average errors are on the order of what the conventional formulas would estimate, whereas individual positional errors can be much smaller or much larger. We added a detailed discussion of this to the Supplementary Note 1.

In our previous version, the errors in the time-delay data were larger than those in the dark structures as the former depended on extrapolation of structure factor amplitudes, which introduces errors. In our revised version, this is no longer the case as all structures (dark and light) are refined against the observed structure factor amplitudes and no extrapolation is performed.

Likewise, why do Extended Data Figures 5c to 5e systematically show lower dark-values for the lowest fluence data (5 mJ/cm²)? If this reflects systematic differences in two dark structures arising from differences in X-ray diffraction data collected in two experiments, then these systematic differences should be encompassed within appropriate coordinate uncertainty estimates. Or, to emphasize the point, the numerical values for the dark structures shown in Extended Data Figures 5c to 5e should agree within coordinate uncertainty. Currently they do not.

We agree that this is unexpected. One reason could be small (differing) amounts of met-myoglobin in one (or both) of the carboxymyoglobin crystal batches (see Vojtechovsky et al., Biophysical Journal, [https://www.cell.com/biophysj/fulltext/S0006-3495\(99\)77056-6](https://www.cell.com/biophysj/fulltext/S0006-3495(99)77056-6)). This is entirely possible and cannot be tested spectroscopically with the accuracy required.

It emphasizes the importance of collecting reference (dark) data of the same crystal batch.

Please check this. Also, please give coordinate error bars in all figures in the manuscript, including in all main figures of the main text. The contribution of both light and dark coordinate uncertainty must be included when differences are shown (eg. Fig. 3a and Extended Data Fig. 3b). Currently those figures do not show coordinate uncertainty.

Where the graphs of the three fluences are shown in a single window, the error bars make the figure extremely cluttered and they become unreadable. For that reason, every figure in which the three traces are shown together has a counterpart in which the three traces are shown separately. For instance, Figure 2 has Extended Data Figure 7 as its counterpart with error bars, and Extended Data Figure 8 has Extended Data Figure 9 as its counterpart with error bars.

Quantum Chemistry

I am not an expert in quantum chemistry calculations and therefore I will not examine critically these calculations. I note that the authors make a connection between quantum chemistry predications and their experimental data when they write:

“Instead, the ~ 300 fs time constant of the apparent increase of CO* occupancy is reminiscent of the damping constant of a coherent nuclear oscillation of CO* that was predicted by recent computational wavepacket analysis¹⁵”

Taking at face-value that oscillations in the CO* occupancy have been predicted with a period of 42 fs, then the 70 ps time-step in the TR-SFX data (Table 1) does not allow these oscillations to be resolved, which the authors acknowledge. Moreover, all experimental data show a time-dependent growth of the CO* occupancy for all pump-laser fluence values (Fig. 2a, left). I note that, for the 5 mJ/cm², the first time-point

appears to have a CO* occupancy of zero, but I suspect that this is a signal-to-noise issue rather than a unique observation for the 5 mJ/cm² data-set. Below a certain occupancy, some features simply cannot be seen in the electron density maps, as reflected in the authors dismissal of their data collected with a pump laser fluence of 2.4 mJ/cm².

There is no doubt that the CO molecule will initially be very hot as it carries off the excess kinetic energy associated with its bond to the haem being broken. This effect could explain the initial disorder of the CO* position without the need to appeal to quantum oscillations. Thus, the apparent CO* occupancy would increase as the CO* molecule cools and becomes more ordered, and this could explain the time-dependence in the data observed above.

The higher the photoexcitation energy, the more energy is deposited into the system, the “hotter” it should be. We relate this fact to the observation that the dynamics becomes much faster in the 101 mJ/cm² data and the magnitude of the structural changes larger. According to this reasoning the

effect described by the referee “the apparent CO* occupancy would increase as the CO* molecule cools and becomes more ordered, and this could explain the time-dependence in the data observed above” should apply to the 101mJ/cm² data and not to the low fluence data. But we observe the opposite.

Therefore, given the possibility of other explanations, [the explanation by the referee is inconsistent with the data] and because the manuscript lacks the time-resolution required to observe coherent oscillations with a period of 42 fs, I feel that the interpretation of these observations as arising from ultrafast quantum oscillations is overstated and may be criticized by others.

What the referee means is that he criticizes this interpretation.

Or, to state it very clearly, if the authors wish to claim evidence of coherent oscillations with a period of 42 fs, then they need to collect TR-SFX data with a time-step of approximately 20 fs. My understanding is that this is beyond what is possible at XFEL sources today but it may become possible in the future.

We agree that it would be highly desirable to visualize the 42 fs oscillation directly. We clearly state that the current data does not allow this, we interpret the apparent CO* occupancy as an ordering of the CO* position and link this to the predicted decay rate of the CO* oscillation. The referee is correct that resolving the 42 fs oscillation by TR-SFX is currently at the limit of what can be done, we have had beamtime proposals at both LCLS and SwissFEL for years to do just this. So far neither facility is stable enough.

Slower oscillatory motions

In some sections of the manuscript, the authors make statements regarding the possibility of biologically relevant collective oscillatory motions, writing for example:

“Moreover, the influence of the photoexcitation regime on oscillatory motions – which are much more pronounced in the low fluence data (5 mJ/cm²) - complicates identification of coherent oscillations that are involved in mode coupling and ultimately result in the biologically relevant structural changes.”

One example is given by Figure 3c of the current manuscript (there are others) which I paste below.

With the figure caption stating:

“Importantly, a strong oscillatory modulation (period of ~ 300 fs) is only visible for the 5 mJ/cm² data.”

Comparing this with the observation of ultrafast collective motions presented in their earlier publication under conditions of multi-photon excitation (Figure 4b of Barends et al. (6), left), I consider that the earlier data more convincingly demonstrate oscillations. I therefore do not accept the author's claim that oscillatory motions are much more pronounced in the low fluence (5 mJ/cm²), single-photon excitation data.

The referee is right that we should have written in the figure legend "Importantly, a strong oscillatory modulation (period of ~ 300 fs) **of the F-helix** is only visible for the 5 mJ/cm² data."

There are lots of oscillations in all data, the statement in Figure 3 referred specifically to the F-helix.

"Biological relevance"

The authors use the phrase "biologically relevant" four times in the main article, including twice in the abstract, and the phrase "nonphysiological pathways" is also used in the abstract. Their point is spelled out quite clearly when they write:

"Valid characterization of biologically relevant single photon-induced reactions can only be made in the regime of linearly increasing signal with pump laser energy."

While I believe that everyone in the field accepts that the photoexcitation conditions used for all time-resolved X-ray diffraction studies of light-sensitive proteins to date risked inducing multiphoton excitation within the crystals to varying extents, there has never been a convincing demonstration of multiphoton induced artefacts in any published difference Fourier electron density map. Brändén & Neutze summarized this status in a recent review (1) when writing:

"Despite these concerns, TR-SFX experiments spanning 10 orders of magnitude in pump laser power density (Table 1) show electron density changes that correlate with functional pathways and concur with biologically important structural changes on microsecond (Fig. 2, C and D) (7, 62) and millisecond (Fig. 2E-H) (54, 68, 70, 71) time scales. These observations suggest that proteins may have evolved to direct all deposited energy toward functional outcomes (53). Although these observations do not prove that multiphoton and single-photon pathways are identical, they highlight the value of working above single-photon excitation thresholds (Table 1) to maximize crystallographic occupancy (Box 3) while remaining within the regime for which the evolution of spectral changes is consistent with single-photon excitation. It is, however, important to repeat ultrafast TR-SFX studies at lower power densities once photoexcitation conditions have been optimized using time-resolved spectroscopy (55, 127)."

Barends et al. explicitly challenge this conclusion with their closing remark (reference 16 is Brändén & Neutze)

"Our results call into question recent statements promulgating the value of TR-SFX pump-probe experiments performed above single-photon excitation thresholds.¹⁶"

Perhaps the authors may consider if they are being unnecessarily confrontational? One aspect is that the photodissociation of carbon monoxide from the active site haem of myoglobin is not a physiological reaction. Rather, it is a model system in biophysics with only indirect biological relevance. Moreover, some scientists question the biological relevance of reactions occurring in crystals, or reactions that occur at a pH that differs from the natural cellular environment, or reactions that are not guided by the crowded environment of a cell etc... There is no universally accepted

definition of “biological relevance”. Rather, there is always a need for judgement based upon the weight of experimental evidence from multiple sources. In my view, the field of time- resolved X-ray crystallography has always been careful to make connections to results from other biophysical studies. More importantly, the authors do not show direct evidence of multiphoton induced artefacts arising in their difference Fourier electron density maps; they do not show direct evidence of multiphoton

induced artefacts arising in their global internal distance matrix comparisons of backbone atom displacements; their analysis of the light-induced displacements of C α atoms relative the porphyrin nitrogen atoms shows a high-level of similarity in the nature of these motions for all pump-laser fluences; and while the authors suggest that there may be differences in the amplitudes of these motions due to multiphoton effects, there appears to be inconsistencies in their estimates of the photoexcited populations using different approaches (Figures 1b, 2a, Table 1) as well as relative to the crystallographic occupancy recovered in their earlier work (6). These concerns recur when considering other motions, such as the amplitude of the out-of-plane distortion of the haem. This highlights the need to draw structural conclusions in a way that does not depend critically upon the accuracy of the estimated crystallographic occupancy, and I have made some suggestions in this direction that utilize the Pearson correlation coefficient. The authors do present evidence that multiphoton excitation may accelerate the protein motions previously described by Barends et al. (6) on the sub-picosecond time-domain (time-constant of 200 to 300 fs). But this specific conclusion makes intuitive sense due to the additional kinetic-energy input into the system by multiphoton absorption, and there is no reason to suspect that this has forced the protein response into “nonphysiological pathways”. Moreover, the strength of this conclusion is weakened by the fact that the first time-point in every data-set is similar to the rate-constant that is being estimated (first time delay $\Delta t = 254$ fs, 5 mJ/cm²; 320 fs, 23 mJ/cm²; 312 fs, 101 mJ/cm²) and therefore the growth of the structural signal could have been better sampled (I appreciate that this was a technical problem and was not in the experimental design, but nevertheless the argument is not really watertight because there is experimental inaccuracy). Irrespectively, no one is suggesting that TR-SFX is superior to spectroscopic methods when measuring reaction rate constants.

In conclusion, the structural results presented in this article show that pump-laser fluence dependent structural differences in the myoglobin:carbon monoxide system are very subtle but may affect the sub-picosecond kinetics. The nature of the protein’s ultrafast structural response is surprisingly independent of pump laser fluence and no signs of damage or side-reactions can be seen in the difference Fourier electron density maps. This very strong conclusion is not clearly acknowledged in the article, and is actually quite close to the conclusions reached earlier by Brändén & Neutze that the authors explicitly challenge in their closing remarks. Looking very hard for multiphoton induced structural differences but failing to find them, despite performing extremely good experiments with extremely good crystallographic data, is a very good result that provides a valuable service to the field. Unfortunately, the authors seem determined to assert that a straightforward chemical and biological interpretation of difference Fourier electron density maps is not possible because of multiphoton excitation. Thus, for the reasons detailed above regarding signal-to-noise issues, uncertainty associated with estimating crystallographic occupancy, and coordinate errors, I do not find the current manuscript convincing. I am disappointed by this, since I have the highest respect for the work of the Heidelberg group and their long-term dedication to high standards in their crystallographic analysis.

We are very grateful for the time spent to referee this manuscript extremely critically. This was extremely valuable and has resulted in a thoroughly revised manuscript that we consider better than the previous version.

We hope to have addressed the concerns with the revised version that contains a reanalysis of the data. The previous inconsistencies caused by different methods to analyze different data sets were removed by consistent analysis. Many of the concerns raised rightfully have been addressed in the new Supplementary Note 1. Nevertheless, the bottom line of our manuscript has not changed. We stand by

our finding that we see significant differences in the dynamics of the structural response and the photophysical mechanism of the dissociation reaction that are related with pump fluence and single vs multiphoton absorption.

Richard Neutze, 07/02/2023.

Other Specific comments:

Table 1a: please give the crystallographic occupancy values.

This has been added as requested.

Table 1c and 1d: Is it really true that exactly 20,000 images have been merged for these data-sets? If this is true, then please footnote the reasons in the table.

Indeed it is; for these data the timing reproducibility did not allow the collection of individual time points as single data sets. Thus, the data were binned, dividing them into different time points according to their actual time delay as determined by the timing tool, into bins of exactly 20,000 images (similar to Barends et al 2015). This is now discussed explicitly in the Materials and Methods Section.

I think that the authors provide a good discussion of some of the challenges in these type of experiments in Supplementary Note 1. In particular the acknowledgement that the scatter from the microjet may not always achieve the idealized case is appreciated.

References cited in this report:

1. G. Branden, R. Neutze, Advances and challenges in time-resolved macromolecular crystallography. *Science* **373**, eaba0954 (2021).
2. V. Srajer *et al.*, Photolysis of the carbon monoxide complex of myoglobin: nanosecond time-resolved crystallography. *Science* **274**, 1726-1729 (1996).
3. R. Henderson, J. K. Moffat, The difference Fourier technique in protein crystallography: errors and their treatment. *Acta Crystallogr. B* **27**, 1414-1420 (1971).
4. M. L. Grunbein *et al.*, Illumination guidelines for ultrafast pump-probe experiments by serial femtosecond crystallography. *Nat Methods* **17**, 681-684 (2020).
5. G. Nass Kovacs *et al.*, Three-dimensional view of ultrafast dynamics in photoexcited bacteriorhodopsin. *Nat Commun* **10**, 3177 (2019).
6. T. R. Barends *et al.*, Direct observation of ultrafast collective motions in CO myoglobin upon ligand dissociation. *Science* **350**, 445-450 (2015).
7. E. De Zitter, N. Coquelle, P. Oeser, T. R. M. Barends, J. P. Colletier, Xtrapol8 enables automatic elucidation of low-occupancy intermediate-states in crystallographic studies. *Commun Biol* **5**, 640 (2022).
8. M. L. Grunbein *et al.*, Effect of X-ray free-electron laser-induced shockwaves on haemoglobin microcrystals delivered in a liquid jet. *Nat Commun* **12**, 1672 (2021).
9. A. Gorel, I. Schlichting, T. R. M. Barends, Discerning best practices in XFEL-based biological crystallography - standards for nonstandard experiments. *IUCrJ* **8**, 532-543 (2021).
10. U. K. Genick, Structure-factor extrapolation using the scalar approximation: theory, applications and limitations. *Acta Crystallogr D Biol Crystallogr* **63**, 1029-1041 (2007).

Report by Referee 3:

We thank the referee for his/her detailed analysis of our work.

Line 404 – 446: The authors state that, “structure factors were extrapolated to full occupancy using the linear extrapolation approximation (Ref 46,47).” However, there is not enough information given so that others working with the data could reproduce with certainty the authors’ analysis in this study. The authors go on to state that, “the methods recently evaluated by de Zitter and coworkers (Ref 53) did not result in appreciable improvements. We therefore did not apply these methods.” One of the strong benefits to the community from the approach of Zitter et al is that the extrapolation approximation analysis is well documented and can be easily reproduced by others. The current manuscript does not meet that standard and more information is required for this type of analysis.

We do not agree. What is required for a scientific publication is that the description is detailed enough that the sample, experiment and analysis can be reproduced. Our manuscripts (the original and revised) version meet this criterion, particularly as we stated that our scripts will all be made available. It is true that XtraPol8 is extremely convenient, but our scripts are not more complicated than an XtraPol8 input file. Moreover, they do not perform any calculations other than what was described in the text (for our original version: scaling with SCALEIT, extrapolation using the exact same formula that was given in the text, refinement with PHENIX, etc.).

In the revised version, however, in response to concerns raised by referee 2, we have completely reanalyzed our data and now use a method that is not (yet) available in XtraPol8. It again exclusively uses programs from the well-documented CCP4 and PHENIX packages for crystallographic calculations, and no structure factor extrapolation is used. The entire process is clearly described and benchmarked in the new Supplementary Note 1. Moreover, as previously, all scripts used will be deposited with the zenodo server, together with the raw data and the results, in order to allow anyone to reproduce the work.

Line 448 – 453: “To obtain error estimates for structural parameters such as bond lengths and ...” The authors should indicate what the estimated coordinate errors are for these studies. This information should also be included in Extended Data Table 1. The plot shown in Figure 1c might also indicate error bars.

See request by referee 2. We have added a detailed analysis of coordinate errors to the supplementary information. In short, the estimated coordinate errors obtained by ML methods or from Rfree are *average* errors over the entire structure. Indeed, using bootstrapping analysis we find very wide distributions of coordinate errors, and while the average of this distribution is comparable to what is found using for instance the well-known Rfree-based Murshudov/Johnson formula. However, for well-defined atoms such as the heme iron, main chain atoms or side chains inside the protein, the errors are much smaller, and for residues on the outside, they are much smaller. Most of the structural changes we show concern such very well-defined atoms (such as the heme iron), and their errors as found using bootstrapping are correspondingly small. We discuss this in a dedicated supplementary note. We consider that the average coordinate errors as estimated traditionally are not useful as a comparison to the graphs we show.

The authors are critical to several other published TR-SFX results. The current work appears to have benefitted from significant SwissFEL beamtime. It might be useful to suggest a minimum approach that others might consider when XFEL beamtime is scarce.

We feel that this manuscript is not the place for such a discussion and are considering a separate

publication addressing this issue. We would advise to establish feasibility of using “reasonable” photoexcitation first. In case of our MbCO, collect a dataset at 10 ps using 5 mJ/cm², collect enough data to obtain reasonable light-dark difference maps. Reduce fluence by a factor of 2, repeat with more data. This will probably be enough for one beamtime. Then apply again for beamtime, now with good justification for the photoexcitation conditions and the beamtime required. Our own experience shows that it is rare to do the experiment correctly during the first time. One tries to achieve too much in too little time with too little knowledge about the system.

Line 71: It is much better to spell out “sub-picosecond” and “nanosecond” rather than use abbreviation with a number before the ps or ns usage.

We agree only partially. It is standard to write ... “a time constant of 100 ps”
But not “a time constant of 100 picoseconds”.

Line 97-99: “Our first power titrations...”

Please state that this is applied to samples in solution.

We thank the referee for this suggestion and have made this correction.

Line 358 “The protein crystals were introduced into the XFEL beam in a thin jet using a gas dynamic virtual (GDVN) nozzle injector (Ref 38).” Please indicate the typical crystal density.

This has been done. In contrast to other groups who report a number density (xy crystals/ml), we report crystal concentrations as percentage (v/v) of gravity settled material. In the current case it was 10 %.

Line 383 “... 20,000 images...” and elsewhere as appropriate.

This should probably read “20,000 indexed lattices” since it is possible that some images may contain more than one lattice.

This has been done. We did not use multi-lattice indexing mode.

Lines 512 – “Figure 1...”

a) Difference electron density...

Please indicate that these are Fobs(light) - Fobs(dark) difference electron density maps and if they are extrapolated maps. Please indicate the resolution of the maps in the legend.

These are Fobs(light) - Fobs(dark) difference density maps. We added the resolution.

The “Fluence” labels may be too small to read when published; please enlarge them.

Please annotate the image to label the electron density feature indicative of CO*. If not too busy, one could also label the feature for the Fe out of plane location. Please label His94 and His63 and Phe43 residues in the image since these are referred to in several other places in the manuscript.

We have increased the label sizes throughout the figure. We added the labels for the CO and CO* and the histidines; there is too little place to label the iron-OOP and Phe43 is almost invisible in this view so we were unable to label this residue. In case our manuscript will be accepted, Nature will reformat the figures anyway to meet their standards.

b) Apparent occupancy of the CO* state...

Please be sure to define CO* in the legend as the photodissociated transient state.

Thanks, this has been done

c) The legend should probably read, “Iron-out-of-plane (OOP-subscript-Fe) distance...” The reader may also wonder what is the uncertainty in the distances shown in the plot; can error bars be added?

We thank the referee; the error bars from bootstrapping have been added

d) $\text{C}\alpha$ - $\text{C}\alpha$ -distance change matrices...

The text numbers along the two axes are too small to read. The axes should include labels such as "residue number"

The "Fluence" labels may be too small to read when published; please enlarge them to match the top.

We have increased the font size of all labels.

Figure 2.

The yellow text on a white background is very low contrast and will be hard to read.

We agree and have changed it to orange; the colors are now in keeping with the IBM Design Library colorblind-friendly palette.

Line 587 "a. Power titration data"

Please indicate that these data were collected with a 10 ps delay time.

This has been changed as requested

Extended Data Table 1 (line 588 -):

"No. images" should probably be the number of lattices.

We changed this to "indexed lattices".

Line 640: "allows identification of the linear photoexcitation regime; it is $\leq 10 \text{ mJ/cm}^2$."

Please add context to this statement.

We have added the required context.

Considering Extended Data Table 2: Laser and excitation parameters, this regime appears to be under an average of one photon per heme (18 mJ/cm^2).

We are not entirely sure what the referee means here.

The average number of absorbed photons is 1.0 /heme.

Line 713 "Extended Data Fig. 6" – Please add residue names to the right-hand illustration.

Thank you for this suggestion, the residue names have been added

G. References: appropriate credit to previous work?

Line 65: It would also be good to add this reference: Chapman, H. N., Annu. Rev. Biochem. 2019. 88:35–58; “X-Ray Free-Electron Lasers for the Structure and Dynamics of Macromolecules”
<https://doi.org/10.1146/annurev-biochem-013118-110744>

This has been added

H. Clarity and context: lucidity of abstract/summary, appropriateness of abstract, introduction and conclusion

The manuscript is well written, the illustrations are appropriate and often clear; however, some of the annotations will be very difficult to read.

Line 30: “However, all ultra-fast TR-SFX studies to date have employed such high pump laser energies that several photons were nominally absorbed per chromophore (Refs 2 -14).”

Some readers may wonder about “all” in this statement. Indeed, the first reference called is Ref 2, (Barends, T.R. et al. Direct observation of ultrafast collective motions in CO myoglobin upon ligand dissociation. Science 350, 445-50 (2015) DOI: 10.1126/science.aac5492). Within Ref 2, these same senior authors state: “The laser energy was set to 5 $\mu\text{J}/\text{pulse}$. This corresponds to \sim one photon/heme given the experimental parameters, the protein concentration in the crystals (51 mM), and an average crystal thickness of 6 μm . (This value has a large variation given the plate-like shape of the crystals and the fact that we had no control over their orientation.) Thus, assuming a 90 μm spot, the power density was 380 GW/cm^2 .”

The referee is right that the statement in Barends et al 2015 is incorrect. It was significantly more than nominally 1 photon/heme

Considering the natural evolution of R&D projects, and e.g. Ref 16 (Brändén, G. & Neutze, R. Advances and challenges in time-resolved macromolecular crystallography. Science 373, eaba0954 (2021), it appears that Barends et al (2015) may have used about five photons/heme for their prior Mb-CO photodissociation studies rather than one photon per heme. The current manuscript should probably very briefly address this explicitly in the abstract and/or introduction.

We state explicitly that Barends et al 2015 used multiphoton excitation. The Nature format is too short to add a discussion on this in the abstract/main text. We added a line to Extended Data Table 2 pointing out that Barends et al 2015 reported an incorrect 1 photon/heme excitation.

Fortunately, this is addressed much more fully in Extended Table 2 and in several other places deeper in the manuscript and supplemental material.

Reviewer Reports on the First Revision:

Referees' comments:

Referee #2 (Remarks to the Author):

Report on: Influence of pump laser fluence on ultrafast dynamics in myoglobin, Barends et al.

The authors have made major changes to their analysis and presentation of their work in a very respectful and careful response to the input of all reviewers. It is impressive that it is possible to achieve such a good signal-to-noise in time-resolved serial femtosecond X-ray crystallography studies of the photodissociation of carbon monoxide from myoglobin with the pump laser fluence in the single-photon domain. I am also very pleased that an apparent inconsistency between this work and an earlier publication by the same team (Barends et al. *Science* 2015) has now been resolved by changing the structural refinement protocol to an ensemble refinement procedure. Moreover, the authors' reflections on the influence of the photoexcited state occupancy on the structural conclusions are now very thorough. The authors are correct that an argument I gave in my previous report, relating to the influence of disorder at high pump-laser fluence, was wrong, with the logic of my previous argument being reversed (my apologies for this mistake).

Overall, these results are very important for the field and these data deserve to be published. I am satisfied that the authors have demonstrated observable differences in the sub-picosecond structural dynamics of myoglobin as a function of the pump laser fluence. The bar that the authors accept every recommendation of a reviewer is too high, since important advances also must leave room for difference of opinion. I therefore recommend that Nature accept the revised manuscript for publication.

Referee #3 (Remarks to the Author):

manuscript number 2022-11-17751A

Barends et al have revised their manuscript extensively and added significant new details in the main text and with additional information in "Supplementary Notes." For instance, supplementary note 1 alone is 22 pages long with several figures to illustrate the concepts. The additional notes are clearly written and outline the methods used to draw their conclusions, which are supported by the data and the descriptions in the manuscript. They have deposited 45 datasets to the PDB that will be released with publication.

Line 76, "... and photodissociation³." Please add these references: Ishigami, I. et al. Detection of a Geminate Photoproduct of Bovine Cytochrome c Oxidase by Time-Resolved Serial Femtosecond Crystallography. *J Am Chem Soc* 145, 22305-22309 (2023).

<https://doi.org/10.1021/jacs.3c07803>; and Shimada, A. et al. A nanosecond time-resolved XFEL analysis of structural changes associated with CO release from cytochrome c oxidase. *Sci Adv* 3, e1603042 (2017). <https://doi.org/10.1126/sciadv.1603042>

The authors have addressed the concerns raised by each of the reviewers. I agree with their statement in the rebuttal letter that the manuscript is very much improved. They have stated that their early work published in 2015 also used more than one photon per heme and even used a "frowny face" footnote in Extended Data Table 2 to address the issue (line 714), which seems appropriate.

Figure 1b; and in Figure S5 a, b, (possibly elsewhere too): The image shows "ensemble refinement" but the main text uses "multi-copy refinement" in about 5 places and 17 places in Supplementary Information, rather than ensemble refinement. Perhaps update the image(s) or indicate in the legend that ensemble refinement is synonymous with multi-copy refinement. Indeed, the application of multi-copy refinement is a very good modification of the manuscript, and as the authors indicate, the strategy could have impact to other ultrafast systems too.

Line 900 - 901: "zenodo.com" should probably be "zenodo.org" searching in Zendo.org with the term "doi 10.5281/zenodo.7341458" did not return any results. Searching Zendo.org and with "10.5281/zenodo/7341458" returned 2,805,066 result(s) found, and viewing all versions showed 1,775 embargoed results, but I could not find data related to this manuscript. Searching with "7341458" did not yield any results. Searching with "Barends" produced 10 results but none were related to Thomas R.M. Barends. Searching with the term "myoglobin" produced 54 results when viewing all versions, but I still could not find the data. Please be sure the link is correct; and if it is embargoed, then the correct link and data is released with publication.

Supplementary Information, page 19

"Moreover, collecting more data instead of increasing the laser energy should be considered to avoid controversy(ref 12)." This is an important sentence that should probably not be buried so deeply in supplementary information. The concept clearly merits inclusion in the main text and could come near the end of the Conclusions section ... of course the difficult part is limited availability of XFEL time.